# CTCF confers local nucleosome resiliency after DNA replication and during mitosis

Nick Owens[1,2†], Thaleia Papadopoulou[1,2†], Nicola Festuccia[1,2], Alexandra Tachtsidi[1,2,3], Inma Gonzalez[1,2], Agnes Dubois[1,2], Sandrine Vandormael-Pournin[1,4], Elphège P Nora[5,6], Benoit G Bruneau[5,6,7], Michel Cohen-Tannoudji[1,4], Pablo Navarro[1,2*]

[1]Epigenomics, Proliferation, and the Identity of Cells, Department of Developmental and Stem Cell Biology, Institut Pasteur, CNRS UMR3738, Paris, France; [2]Equipe Labellisée LIGUE Contre le Cancer, Paris, France; [3]Sorbonne Université, Collège Doctoral, Paris, France; [4]Early Mammalian Development and Stem Cell Biology, Department of Developmental and Stem Cell Biology, Institut Pasteur, CNRS UMR 3738, Paris, France; [5]Gladstone Institutes, San Francisco, United States; [6]Cardiovascular Research Institute, University of California, San Francisco, San Francisco, United States; [7]Department of Pediatrics, University of California, San Francisco, San Francisco, United States

**Abstract** The access of Transcription Factors (TFs) to their cognate DNA binding motifs requires a precise control over nucleosome positioning. This is especially important following DNA replication and during mitosis, both resulting in profound changes in nucleosome organization over TF binding regions. Using mouse Embryonic Stem (ES) cells, we show that the TF CTCF displaces nucleosomes from its binding site and locally organizes large and phased nucleosomal arrays, not only in interphase steady-state but also immediately after replication and during mitosis. Correlative analyses suggest this is associated with fast gene reactivation following replication and mitosis. While regions bound by other TFs (Oct4/Sox2), display major rearrangement, the post-replication and mitotic nucleosome positioning activity of CTCF is not unique: Esrrb binding regions are also characterized by persistent nucleosome positioning. Therefore, selected TFs such as CTCF and Esrrb act as resilient TFs governing the inheritance of nucleosome positioning at regulatory regions throughout the cell-cycle.
DOI: https://doi.org/10.7554/eLife.47898.001

*For correspondence:
pablo.navarro-gil@pasteur.fr

†These authors contributed equally to this work

## Introduction

Gene regulatory processes are frequently governed by sequence-specific Transcription Factors (TFs) that recognize specific DNA binding motifs enriched at promoters and enhancers (*Spitz and Furlong, 2012*). To gain access to DNA, which in eukaryotes is wrapped around a histone core octamer – the nucleosome (*Luger et al., 1997*) – TFs employ different strategies (*Voss and Hager, 2014*). Whereas nucleosomal DNA is accessible to pioneer TFs (*Cirillo, 1998*; *Zaret and Carroll, 2011*), other TFs require co-factors to evict or displace nucleosomes that occlude their binding sites. These co-factors can be additional TFs that act cooperatively to destabilize nucleosomes (*Mirny, 2010*; *Miller and Widom, 2003*), and/or ATP-dependent activities that slide or evict nucleosomes and thus remodel cis-regulatory elements (*Becker and Workman, 2013*; *Li et al., 2014*). Typically, TF binding regions are characterized by Nucleosome Depleted Regions (NDRs) centered on TF motifs and flanked by Nucleosome Ordered Arrays (NOAs) (*He et al., 2010*; *Wang et al., 2012*; *Valouev et al., 2011*; *Teif et al., 2012*). This is particularly well illustrated by the genomic binding sites of the zinc

**eLife digest** A single cell contains several meters of DNA which must be tightly packaged to fit inside. Typically, the DNA is wound around proteins, like a thread around many spools, to form more compact structures called nucleosomes. Before a cell divides in two, however, it needs first to access and replicate its DNA so that each new cell can get a copy of the genetic material. The cell then needs to condense the DNA again so that the two copies can be easily separated via a process called mitosis. These two processes – DNA replication and mitosis – entail major rearrangements of the nucleosomes, which then need to be returned to their original positions.

Nucleosomes are also repositioned when cells need to access the coded instructions written in genes. Molecules called transcription factors bind to targets within the DNA to make sure genes are active or inactive at the right times of a cell's life, but many are evicted from the DNA during its replication and during cell division. Most transcription factors also require nucleosomes to be specifically organized to bind to the DNA, and it remains unclear how the factors re-engage with the DNA and how nucleosomes are managed during and after DNA replication and mitosis.

Owens, Papadopoulou et al. set out to understand how nucleosomes are organized immediately after DNA is replicated and while cells divide. Experiments with mouse cells grown in the laboratory showed that certain transcription factors can rebind to their targets within minutes of replication finishing, remain bound to the DNA during cell division, and displace nucleosomes from their binding sites. Owens, Papadopoulou et al. refer to these factors as "resilient transcription factors" and identified two examples, named CTCF and Esrrb. Further experiments showed that, by maintaining the structure of nearby nucleosomes while a cell divides, these resilient transcription factors could quickly reactivate genes immediately after DNA replication and mitosis are complete.

These findings show that transcription factors play a fundamental role in maintaining gene regulation from one generation of cells to the next. Further studies on this topic may eventually foster progress in research areas where cell division is paramount, such as regenerative medicine and cancer biology.

DOI: https://doi.org/10.7554/eLife.47898.002

finger CCCTC-binding protein (CTCF) (*Fu et al., 2008*; *Teif et al., 2014*), a TF involved in chromatin organization and transcriptional control (*Nora et al., 2017*; *Merkenschlager and Nora, 2016*). However, the passage of the replication fork during DNA replication and the condensation of the chromatin during mitosis are associated with a general loss of TF binding and nucleosome positioning at their target sites (*Ramachandran and Henikoff, 2016*; *Festuccia et al., 2019*). Whether TFs are readily required to either maintain or rapidly rebuild local nucleosome architectures during or after replication and mitosis remains unclear.

DNA replication leads to a period during which TFs and nucleosomes enter into direct competition; in *Drosophila* S2 cells, the reconstitution of specific NDRs/NOAs over active regulatory elements, particularly at enhancers, takes much longer than previously anticipated (*Ramachandran and Henikoff, 2016*). Similarly, in mouse Embryonic Stem (ES) cells, chromatin accessibility over TF binding sites is lost during replication and progressively reacquired as nascent chromatin matures (*Stewart-Morgan et al., 2019*). During mitosis, regulatory elements display strongly attenuated nucleosome phasing and, more strikingly, enhancers are invaded by stable nucleosomes, as shown in ES cells (*Festuccia et al., 2019*). Hence, both replication and mitosis can be seen as a *tabula rasa* of functional interactions between TFs, their cognate motifs and local nucleosomal architectures. Thus, how proliferating cells maintain or restructure nucleosome arrays over regulatory elements as they undergo cycles of replication and mitosis, is largely unknown. This seems particularly important during early development, when TFs not only instruct but also maintain cell identity (*Soufi and Dalton, 2016*; *Festuccia et al., 2017a*; *Festuccia et al., 2017b*; *Egli et al., 2008*). For instance, the TF Zelda was shown to be continuously required during early *Drosophila* development, suggesting that by means of its pioneering activity it is capable of rapidly rebinding its targets after the passage of the replication fork (*McDaniel et al., 2019*). While direct, nucleosome-based evidence is still lacking, it is likely that Zelda ensures the rapid reestablishment of NDRs/NOAs at its binding sites after replication (*McDaniel et al., 2019*). Moreover, recent evidence does not favor a model in which Zelda

directly controls its target sites during mitosis (*Dufourt et al., 2018*). In contrast, the TF Esrrb was shown to act as a mitotic bookmarking factor that binds thousands of regulatory elements in mitotic ES cells (*Festuccia et al., 2016*). At these sites, the nucleosomes preserve an interphase-like configuration whereas at regions losing TF binding nucleosomal arrays are largely disorganized (*Festuccia et al., 2019*). Whether Esrrb also maintains nucleosome positioning during replication remains however unknown.

The incomplete correlations that are currently available suggest a model in which specific TFs may govern nucleosome positioning during replication and/or mitosis, a mechanism that can potentially complement the inheritance of gene regulatory states by independent epigenetic mechanisms. Here, we focus on CTCF to show that this TF is strictly required to maintain nucleosome positioning in interphase, immediately after replication and during mitosis, in mouse ES cells. While this is also observed at Esrrb binding regions, those bound by other TFs such as Oct4/Sox2 display significant nucleosome rearrangement. Further, we show that genes rapidly reactivated after replication and mitosis are closely associated with CTCF binding. Therefore, certain, but not all TFs, govern nucleosome positioning and confer chromatin resiliency during replication and mitosis to foster the appropriate re-establishment of transcription profiles.

## Results

### CTCF binds at highly ordered nucleosome arrays in interphase

CTCF binding has been previously shown to take place over large and phased nucleosomal arrays displaying a prominent NDR (*Fu et al., 2008*; *Teif et al., 2014*; *Barisic et al., 2019*; *Wiechens et al., 2016*). Therefore, we established the repertoire of CTCF binding regions in ES cells (cultured in serum and LIF) by Chromatin Immunoprecipitation followed by sequencing (ChIP-seq; *Figure 1A*; top panel; *Supplementary file 1*). We identified 52,129 regions displaying robust peaks that significantly overlap with previous studies (*Nora et al., 2017*; *Davis et al., 2018*; *Pękowska et al., 2018*)

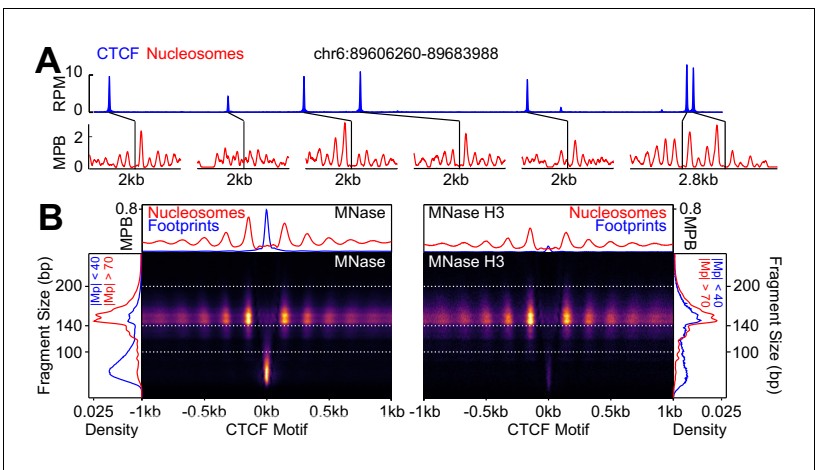

**Figure 1.** CTCF binding and nucleosome positioning. (**A**) Representative genome snapshot (chr6:89606260–89683988; 78 kb) showing CTCF binding in blue (reads per million; RPM) and the associated NDR/NOAs in red (MPB). (**B**) MNase-seq (left) and MNase H3 ChIP-seq (right) V-plots (MNase fragment mid-point vs MNase fragment length) at CTCF binding sites showing +/- 1 kb surrounding CTCF motifs for fragment sizes in the range 30–250 bp. Sidebars indicate densities for fragments with midpoints (Mp) located either within 40 bp of the CTCF motif (highlighting CTCF footprints; blue), or at more than 70 bp (highlighting nucleosomal fragments; red). Top bar gives metaplots of footprints (fragment length <100 bp; blue) and nucleosomes (fragment length within 140–200 bp; red). The Y-axis represent fragment midpoints-per-billion (MPB).

DOI: https://doi.org/10.7554/eLife.47898.003

The following figure supplement is available for figure 1:

**Figure supplement 1.** Additional information on CTCF binding in interphase.

DOI: https://doi.org/10.7554/eLife.47898.004

(*Figure 1—figure supplement 1A*; *Supplementary file 2*). At these regions, we observed that the aggregate of CTCF motifs directly correlates with CTCF occupancy (*Figure 1—figure supplement 1B*), indicating that the presence of its cognate binding sequence drives CTCF recruitment. Next, we used Microccocal Nuclease digestion (MNase-seq), histone H3 ChIP-seq following MNase digestion and Assay for Transposase Accessible Chromatin (ATAC-seq) to profile nucleosomes genome-wide (*Festuccia et al., 2019*). At individual CTCF binding regions, we observed highly organized nucleosome arrays, particularly over large CTCF peaks (*Figure 1A*; bottom panels). To obtain a global picture of how nucleosomes are organized around CTCF binding regions, we generated V-plots (*Kent et al., 2011*; *Henikoff et al., 2011*) centered on the best CTCF motif of each binding region. We observed a very well defined NOA/NDR/NOA structure surrounding CTCF motifs (*Figure 1B* and *Figure 1—figure supplement 1C*). CTCF binding footprints, identified as small MNase (<100 bp) and ATAC (<150 bp) fragments, were also detected within the NDR except upon H3 ChIP (*Figure 1B* and *Figure 1—figure supplement 1C*). These observations indicate that CTCF binding at its motif is closely associated with nucleosome positioning.

## CTCF binding drives local nucleosome positioning in interphase

Next, we explored the relationships between the magnitude of CTCF binding, the presence of its cognate motif and nucleosome positioning. After ranking CTCF binding regions by their peak height, we confirmed that the aggregate of CTCF motifs is correlated with CTCF occupancy (*Figure 2A*). Further, as the motif score and CTCF binding diminishes, the associated MNase- and ATAC-footprints decrease (*Figure 2B*), providing independent validation of the direct correlation between CTCF binding and the presence of its motif. Notably, the reduction of CTCF binding across the regions correlates with a progressive loss of nucleosome order (*Figure 2C*), consistent with what has been previously observed (*Vainshtein et al., 2017*). This is particularly well illustrated by the quantitative analysis of the position of the −1 and +1 nucleosomes, which roll inwards, shrinking the NDR and resulting in increased nucleosome occupancy at the CTCF motif (*Figure 2D*). These observations argue that the interaction of CTCF with its cognate motif acts as a major force driving the establishment of NDRs/NOAs, as previously suggested by siRNA knock-down of CTCF (*Wiechens et al., 2016*). To further establish whether CTCF is readily required to maintain local NDRs/NOAs, we used an Auxin-inducible depletion strategy enabling assessment of the immediate

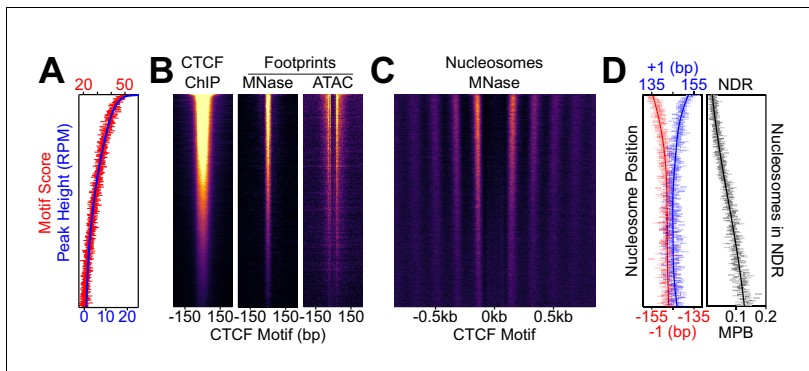

**Figure 2.** Nucleosome positioning correlates with CTCF occupancy. CTCF binding regions were ordered by peak height descending in 100-region bins in all panels. (**A**) Overlaid of the aggregate of motif scores beneath each CTCF binding region (red) and the height of the corresponding CTCF peaks (blue). (**B**) CTCF motif-centered heatmaps for CTCF ChIP-seq and MNase/ATAC footprints. ChIP marks the midpoint of inferred ChIP fragments; MNase marks +/- 4 bp of midpoint for 1–100 bp fragments; ATAC marks cut sites of 1–150 bp fragments shifted inwards by 4 bp. (**C**) Heatmaps of nucleosome-sized MNase fragment midpoints (140–200 bp). (**D**) Quantitative analysis of the NDR. Left; the position (in bp) of the median nucleosomal signal (140–200 bp MNase-seq fragments with midpoints within +/- 70–230 bp from the motif) for the +/- 1 nucleosome (blue and red, respectively) per bin and smoothed with Gaussian process regression (line). Right; mean depth of nucleosomal fragments (MNase H3 ChIP-seq MPB; black) with midpoints in the NDR defined as +/- 80 bp centered on the motif for each bin and then smoothed with a Gaussian process regression.

DOI: https://doi.org/10.7554/eLife.47898.005

consequences of a loss of CTCF. Hence, we exploited ES cells expressing CTCF fused to an Auxin-Inducible Degron (*Nora et al., 2017*) (AID; thereafter CTCF-aid). In accord with the hypomorphic behavior of this line (*Nora et al., 2017*), CTCF-aid displayed reduced expression (*Figure 3—figure supplement 1A*), bulk chromatin association (*Figure 3—figure supplement 1B*) and binding levels (*Figure 3A*) that correlated with less prominently positioned nucleosomes compared to wild-type cells (*Figure 3B*). Upon a short (2 hr) treatment with the Auxin analog Indole-3-Acetic Acid (IAA), CTCF-aid expression (*Figure 3—figure supplement 1A*), chromatin association (*Figure 3—figure supplement 1B*) and binding (*Figure 3A*) were significantly reduced. This led to a dramatic loss of NOAs, major displacements of the +/- 1 nucleosomes, and an invasion of the NDR by nucleosomes (*Figure 3B,C*). Since this effect was observed across different classes of functional genetic features (*Figure 3D*), and rapidly follows loss of CTCF, we conclude that CTCF is a major determinant of local nucleosome positioning in steady-state conditions.

## Replication alters nucleosome positioning at enhancers but not at CTCF-bound regions

During replication, TFs are evicted from their targets and the chromatin has to be reconstituted downstream of the replication fork (*Ramachandran and Henikoff, 2016*; *Groth et al., 2007*; *Stewart-Morgan et al., 2019*). Hence, after replication, TFs and nucleosomes have been shown to be in direct competition, a phenomenon that substantially delays the reconstitution of proper NDRs/

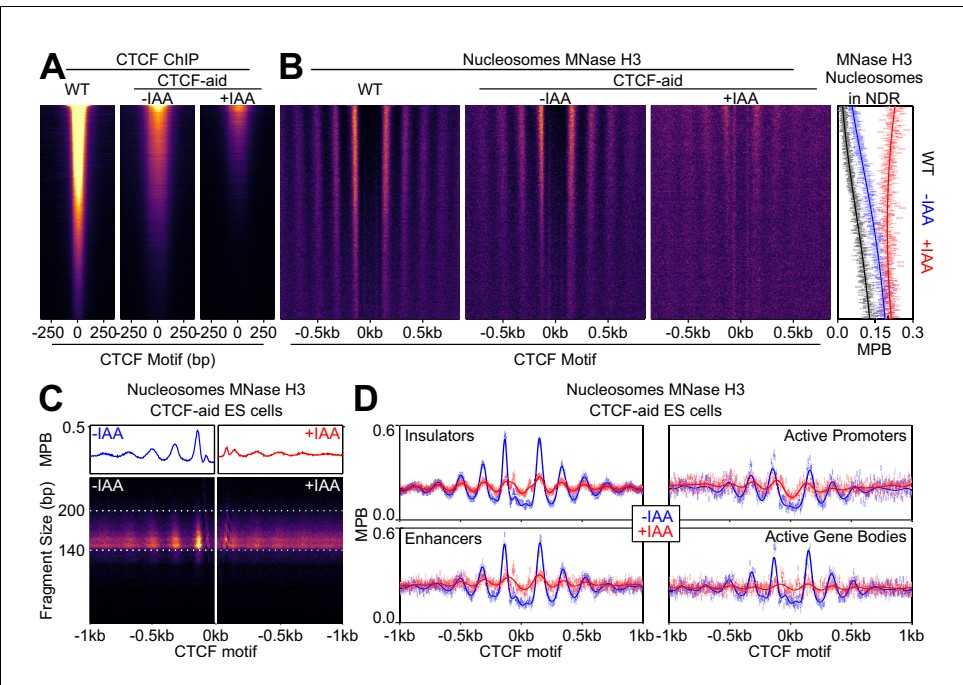

**Figure 3.** CTCF drives local nucleosome positioning. (**A**) CTCF ChIP-seq in wild-type (E14Tg2a; WT) and CTCF-aid -/+ IAA (2 hr treatment) ES cells, presented as in *Figure 2B* and scaled to WT. (**B**) Nucleosome analysis as a function of CTCF binding. Left; heatmaps of nucleosomal fragments of MNase H3 ChIP-seq in WT and CTCF-aid -/+ IAA presented as in *Figure 2C* and scaled to WT. Right; nucleosomal signal (MBP) within the NDR (+/- 80 bp of motif) for WT (black) and CTCF-aid -/+IAA (blue and red, respectively), presented as in *Figure 2D*. (**C**) Split V-plot and corresponding metaplot of MNase H3 nucleosomal ChIP-seq signal presented as in *Figure 1B* for -IAA (left) and +IAA (right). (**D**) Metaplots of MNase H3 ChIP-seq for -IAA and +IAA centered at CTCF motifs of CTCF peaks intersecting with the indicated ChromHMM categories. Datapoints mark mean MPB per site at each base pair; lines represent Gaussian process regression.

DOI: https://doi.org/10.7554/eLife.47898.006

The following figure supplement is available for figure 3:

**Figure supplement 1.** Auxin-induced degradation of CTCF-aid in interphase and mitosis.

DOI: https://doi.org/10.7554/eLife.47898.007

NOAs over enhancers, as shown in *Drosophila* S2 cells (*Ramachandran and Henikoff, 2016*). Given that proteomic approaches have identified CTCF on newly synthesized chromatin (*Alabert et al., 2014*), we aimed at studying the consequences of replication over CTCF binding sites. To do this, we used Mapping In vivo Nascent Chromatin with EdU (MINCE-seq), a technique that enables the capture of newly synthetized nucleosomal DNA following MNase digestion (*Ramachandran and Henikoff, 2016*). We performed two measurements, one immediately after a short pulse of EdU (2.5 min pulse) and another 1 hr after a chase period without further EdU incorporation (*Figure 4—figure supplement 1A*; *Supplementary file 1*); these two time-points reflect the status of newly replicated (pulse) and maturing chromatin (chase). In contrast to S2 cells, which even 1 hr post-replication still display altered nucleosomal structures at enhancers (*Ramachandran and Henikoff, 2016*), CTCF binding regions display a remarkable nucleosomal resiliency in ES cells. Indeed, only minor changes, if any, were appreciable just after replication (*Figure 4A*; center panel). During the following hour, CTCF binding regions acquire a nucleosomal structure indistinguishable from the controls (*Figure 4A*; right panel). Given the direct role of CTCF in local nucleosome positioning (*Figures 2* and *3*), it is therefore likely that CTCF is capable of rapidly rebinding its sites post-replication to efficiently re-establish NDRs and NOAs. Prompted by the fast reorganization of CTCF-binding regions in ES cells, we explored the post-replication nucleosome dynamics over ES cell enhancers, centered on p300 summit. We observed an increase in nucleosome density over the NDRs and attenuated NOAs immediately after replication (*Figure 4A*; center panel), contrasting markedly with CTCF binding regions. One hour after replication, however, we observed that ES cell enhancers were near completely restored (*Figure 4A*; right panel), indicating that ES cells are more efficient than S2 cells in reconfiguring nucleosome positioning at enhancers.

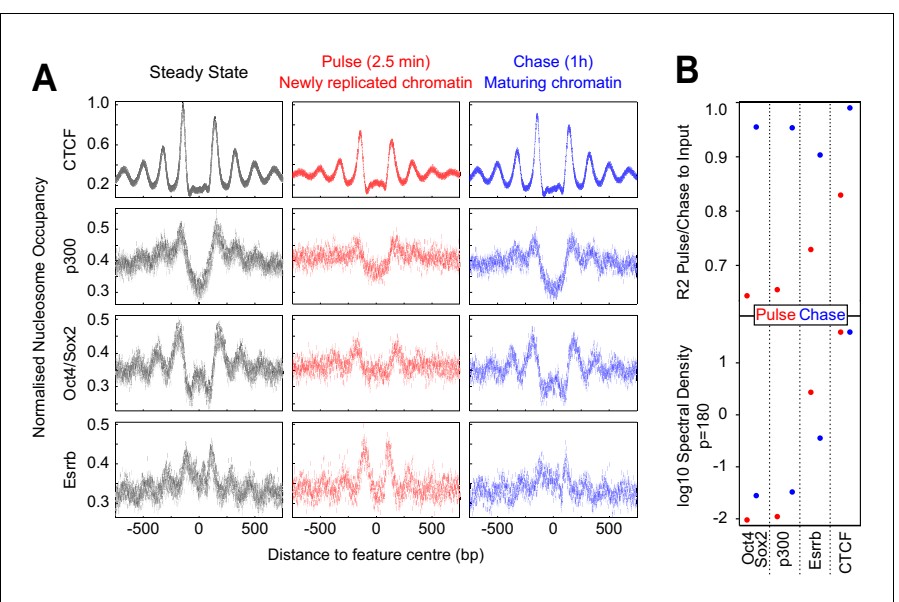

**Figure 4.** Fast nucleosome positioning at CTCF and Esrrb binding regions after replication. (**A**) MINCE-seq metaplots of nucleosomal fragments at CTCF (motif centered), p300 (summit centered), Oct4/Sox2 (motif centered) and Esrrb (motif centered) at steady state (pulse and chase inputs overlaid), pulse (newly replicated chromatin; 2.5 min) and chase (maturing chromatin; 1 hr). Datapoints represent mean MPB per site at each base pair normalized to input/steady-state control. (**B**) Quantification of reconstitution of steady-state nucleosomal order. Top: $R^2$ between relevant pulse/chase and controls. Bottom: $\log_{10}$ spectral density at p=180 (nucleosome + linker) of Gaussian process squared exponential covariance function, with optimized hyperparameters.
DOI: https://doi.org/10.7554/eLife.47898.008

The following figure supplement is available for figure 4:

**Figure supplement 1.** MINCE-seq controls and validation.
DOI: https://doi.org/10.7554/eLife.47898.009

## Distinct post-replication nucleosome dynamics at regions bound by different TFs

ES cell enhancers are bound by several TFs, often in complex combinations and stoichiometries, as exemplified by regions binding Esrrb, Oct4 and Sox2 (*Heurtier et al., 2019*), three master TFs of pluripotency. At some of these regions nucleosome positioning is organized around the Esrrb motif, and at others, around the Oct4/Sox2 composite motif (*Festuccia et al., 2019*). Thus, we explored whether the dynamics of nucleosome repositioning after replication are distinct over these regions. At regions bound by Oct4/Sox2, nucleosome positioning was altered upon replication (*Figure 4A*). This indicates that the pioneer activity of these two factors (*Soufi et al., 2015*), illustrated here by the detection of a nucleosome overlapping their motif (*Figure 4A*), is not sufficient to rapidly reorganize nucleosomal arrays after replication. Regions organized by Esrrb, however, displayed a prominent NDR centered on the Esrrb motif, two particularly well positioned flanking nucleosomes and a persistent NOA, just after replication (*Figure 4A*; center panel). Strikingly, the NDR and the positioning of the +/- 1 and surrounding nucleosomes were more prominent immediately after replication than after 1 hr (*Figure 4A*). This suggests that following replication, Esrrb is rapidly rebound at these sites and imposes strong nucleosome positioning, which is subsequently modified by the binding of additional TFs – a phenomenon that we previously described when we compared Esrrb binding in interphase and mitosis (*Festuccia et al., 2019*). These observations, together with the analysis of p300-centered enhancers and CTCF binding regions, suggest that different sets of regions substantially differ in post-replication nucleosome dynamics. To obtain quantitative comparisons between these regions, we computed two parameters: the $R^2$ coefficient revealing the similarity of the nucleosome profiles after replication with the controls (*Figure 4B*; top panel) and the spectral density assessing nucleosome periodicity (*Figure 4B*; bottom panel). Both measurements clearly revealed a comparably delayed restoration of nucleosome positioning at p300-enhancers and Oct4/Sox2 regions after replication. In contrast, Esrrb and CTCF, display, in turn, increasing capacity to reinstate NDRs/NOAs minutes after the passage of the replication fork. These differential post-replication nucleosome dynamics among TF binding regions are independently supported by our analysis of repli-ATAC data recently generated in ES cells (*Figure 4—figure supplement 1B*), a technique where chromatin accessibility is measured as chromatin matures after replication (*Stewart-Morgan et al., 2019*).

## CTCF, like Esrrb, is a mitotic bookmarking factor in ES cells

Given that both Esrrb and CTCF can re-establish nucleosome positioning following replication, we then turned our attention to mitosis, where Esrrb has already been shown to bind at sites preserving NDRs/NOAs (*Festuccia et al., 2016*; *Festuccia et al., 2019*). Using MNase-seq data, we observed the typical NDR/NOA structure at CTCF binding regions in mitotic cells (*Figure 5A*) obtained by nocodazole shake-off with >95% purity (*Festuccia et al., 2019*). Therefore, we hypothesized that CTCF acts as a mitotic bookmarking factor (*Shen et al., 2015*; *Sekiya et al., 2017*; *Zhang, 2019*; *Burke et al., 2005*). Thus, we performed CTCF ChIP-seq and observed site-specific interactions in mitotic cells associated with ordered nucleosomal arrays (*Figure 5B*; *Supplementary file 1*). Using a generalized linear model, we identified 29,539 bookmarked sites (56.7%) and categorized them according to the relative sizes of interphase and mitosis peaks (*Figure 5—figure supplement 1A–C*; *Supplementary file 2*). Cohesin, a recurrent partner of CTCF in interphase (*Merkenschlager and Nora, 2016*), was found fully evicted from its targets in mitosis (*Warren et al., 2000*; *Waizenegger et al., 2000*) (*Figure 5B* and *Figure 5—figure supplement 2A*; *Supplementary file 2*), underscoring the specificity of our observation for CTCF. Notably, we observed that nearly all regions displaying robust enrichment for Cohesin in interphase are bookmarked by CTCF, even though Cohesin accumulation is not a prerequisite for CTCF bookmarking (*Figure 5—figure supplement 2B,C*). Moreover, the existence of high quality CTCF motifs appeared to be a better indicator of binding in mitosis than in interphase (*Figure 5—figure supplement 3*), suggesting that the conditions for CTCF binding are more stringent in mitosis. Our finding of CTCF bookmarking activity in ES cells contradicts recent data in human somatic cells (*Oomen et al., 2019*). However, our analyses in mouse somatic cell lines (NIH3T3 and C2C12) revealed no or reduced evidence of mitotic bookmarking activity by CTCF (*Figure 5C* and *Figure 5—figure supplement 4*; *Supplementary file 1*). Therefore, even though CTCF can decorate mitotic chromosomes globally, as revealed by

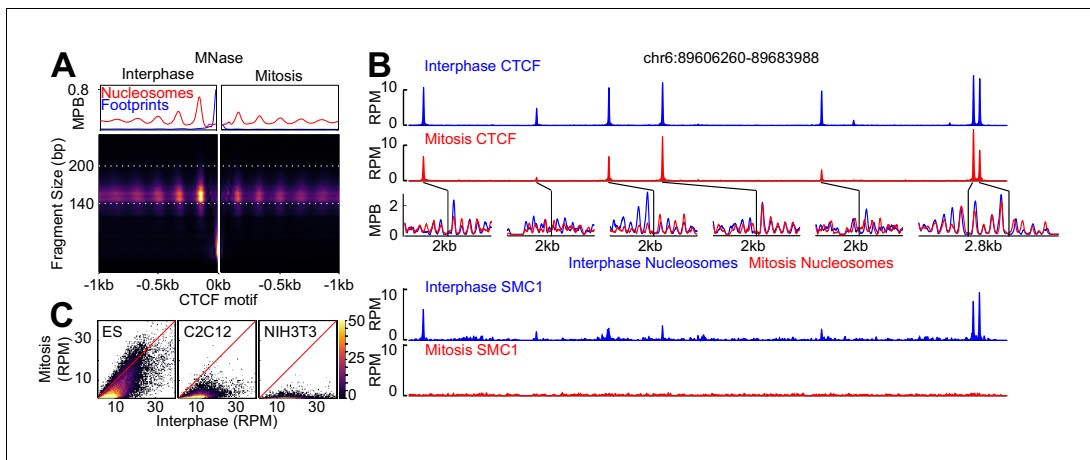

**Figure 5.** CTCF is a mitotic bookmarking factor in ES cells. (**A**) Split V-plot of MNase data at CTCF binding sites in interphase (left) and mitosis (right) presented as in *Figure 3C*. (**B**) Representative genome snapshot (chr6:89606260–89683988; 78 kb; as in *Figure 1A*) for CTCF ChIP-seq (RPM), MNase-seq nucleosome fragments (MPB) and Cohesin (SMC1) ChIP-seq (RPM), in interphase (blue) and mitosis (red). (**C**) Heatmaps of CTCF ChIP-seq signal (RPM per peak) in interphase (X-axis) and mitosis (Y-axis) for ES, C2C12 and NIH3T3 cells, at all peaks identified in ES cells.

DOI: https://doi.org/10.7554/eLife.47898.010

The following figure supplements are available for figure 5:

**Figure supplement 1.** Additional information on CTCF binding in mitosis.
DOI: https://doi.org/10.7554/eLife.47898.011
**Figure supplement 2.** SMC1 binding in interphase and mitosis.
DOI: https://doi.org/10.7554/eLife.47898.012
**Figure supplement 3.** CTCF binding motifs and mitotic bookmarking.
DOI: https://doi.org/10.7554/eLife.47898.013
**Figure supplement 4.** CTCF binding in C2C12 and NIH3T3.
DOI: https://doi.org/10.7554/eLife.47898.014
**Figure supplement 5.** Global behavior of CTCF in mitotic cells.
DOI: https://doi.org/10.7554/eLife.47898.015

microscopy in cell lines and embryos (*Figure 5—figure supplement 5*), this is not necessarily translated into mitotic bookmarking capacity in every cell type.

## Mitotic CTCF binding preserves ordered nucleosomal arrays

We further validated mitotic CTCF bookmarking by analyzing MNase and ATAC footprints after ranking the regions by the magnitude of mitotic binding determined by ChIP-seq (*Figure 6A*). Examining the association between CTCF binding and nucleosome positioning in mitosis, and in complement to the analysis in interphase (*Figure 2*), we found that the progressive reduction of mitotic CTCF binding correlates with a gradual loss of NDRs and NOAs (*Figure 6A*). Moreover, when CTCF binding regions were split as bookmarked or lost in mitosis, we could confirm that the nucleosomes are less well positioned (*Figure 6B,C* and *Figure 5—figure supplement 1D*) and exhibit clear displacements toward the motif (*Figure 6C,D*). Overall, this establishes that CTCF is a mitotic bookmarking TF that preserves nucleosome order during mitosis. Nevertheless, even CTCF bookmarked regions presented a strong displacement inwards of the +1 nucleosome (25 bp for +1 versus 3 bp for −1 nucleosome; *Figure 6A,D*) and a more moderate shift of all following nucleosomes (*Figure 6D*). This indicates that the constraints imposed on the nucleosomes by CTCF are slightly different in interphase and in mitosis. Since in interphase CTCF sites that do not bind Cohesin do not show this repositioning of the +1 nucleosome (*Figure 5—figure supplement 2D*), it cannot be solely explained by the mitotic loss of Cohesin. Additional factors therefore influence nucleosome positioning at CTCF binding regions in either interphase or mitosis. Next, we aimed at exploiting CTCF-aid ES cells to address the impact of losing mitotic CTCF bookmarking directly. We observed

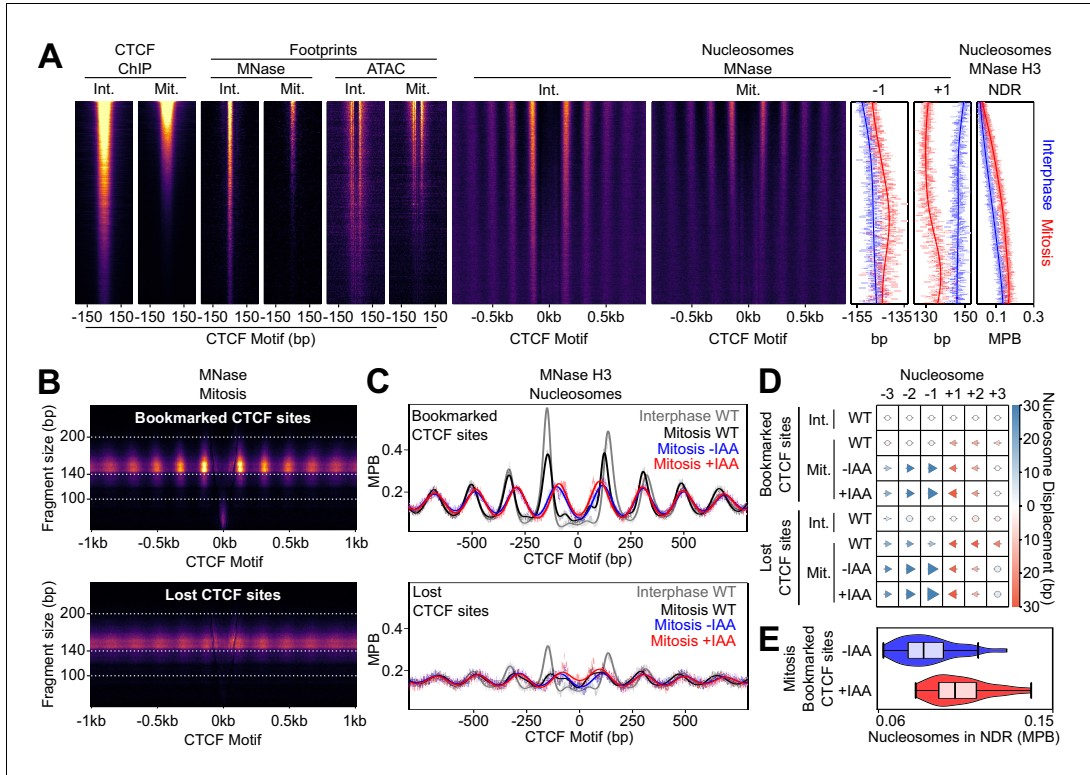

**Figure 6.** CTCF Positions Nucleosomes in Mitosis. (**A**) Interphase (Int.) and mitosis (Mit.) data presented as in *Figure 2* for CTCF ChIP-seq; MNase/ATAC-seq footprints; MNase-seq nucleosomal signal; +/- 1 nucleosome positions and NDR signal (MNase H3 ChIP-seq). The regions are ordered by descending mitotic ChIP-seq peak height. All heatmaps are scaled to interphase. (**B**) Mitosis MNase V-plots for CTCF Bookmarked (top) and Lost regions (bottom), presented as in *Figure 1B*. (**C**) MNase H3 ChIP-seq metaplots at CTCF Bookmarked (top) and Lost sites (bottom) centered on CTCF motifs for wild-type interphase and mitosis (gray and black, respectively) and for mitosis CTCF-aid -/+ IAA (blue and red, respectively), normalized to WT interphase, unnormalized metaplots shown in *Figure 6—figure supplement 1B*. (**D**) Mean +/- 1,2,3 nucleosome positions relative to wild-type interphase, for datasets shown in (**C**), complete unnormalized datasets shown in *Figure 6—figure supplement 1D*. Circles denote nucleosome movements < 5 bp; arrow direction, size and color describes movements > 5 bp. (**E**) Nucleosome signal in NDR ([−40, 40]bp of the motif) at Bookmarked sites in mitosis CTCF-aid -/+ IAA, normalized to WT interphase; unnormalized data shown in *Figure 6—figure supplement 1C*.
DOI: https://doi.org/10.7554/eLife.47898.016

The following figure supplement is available for figure 6:

**Figure supplement 1.** Additional information on CTCF binding and nucleosome positioning in CTCF-aid ES cells.
DOI: https://doi.org/10.7554/eLife.47898.017

that the hypomorphic nature of CTCF-aid binding was amplified in mitosis; CTCF-aid was barely detectable in bulk mitotic chromatin (*Figure 3—figure supplement 1B*) and more specifically at regions binding CTCF in wild-type mitotic cells (*Figure 6—figure supplement 1A*). This global loss of mitotic bookmarking was associated with the acquisition of nucleosomal properties characteristic of regions losing mitotic CTCF binding in wild-type cells: nucleosome positioning was strongly attenuated (*Figure 6C* and *Figure 6—figure supplement 1B*); the NDRs were partially invaded by nucleosomes (*Figure 6—figure supplement 1C*); nucleosomes, especially upstream of the motif, shifted inwards (~55 bp upstream versus 21 bp downstream; *Figure 6D* and *Figure 6—figure supplement 1D*). These changes in nucleosome positioning were only minimally increased upon IAA treatment (*Figure 6—figure supplement 1B–D*), which leads to further invasion of the NDR by nucleosomes (*Figure 6E*). We conclude, therefore, that CTCF behaves as a canonical bookmarking factor in ES cells and actively maintains nucleosome order, mirroring our previous data on Esrrb (*Festuccia et al., 2019*).

## CTCF binding and gene reactivation following mitosis and replication

Gene transcription is strongly inhibited during replication and mitosis, and therefore, genes need to be reactivated in an appropriate manner following these processes. We explored two available datasets to study a potential link between CTCF binding and gene reactivation after mitosis (*Teves et al., 2018*) and replication (*Stewart-Morgan et al., 2019*). For mitosis, we focused on chromatin associated RNAs, enriched in nascent pre-mRNA, in mitotic ES cells and upon release in interphase measured by RNA-seq (*Teves et al., 2018*). We used k-means clustering to find 5 groups of genes displaying different reactivation dynamics, measured as the rate of recovery to the level of pre-mRNA detected in bulk asynchronous cells (*Figure 7A*). Next, we computed the Fisher Exact enrichment of bookmarked and lost CTCF peaks at increasing distances from the promoters of genes in the five clusters (*Figure 7B*). We observed that the two clusters displaying slow reactivation dynamics (Clusters 1 and 2) did not show any enrichment for CTCF binding. Contrastingly, clusters with faster dynamics (Clusters 3, 4 and 5) showed significant CTCF enrichment. While the enrichment at Cluster three was similarly low and distal for both bookmarked and lost CTCF sites, at Clusters 4 and 5 more proximal (<100 kb) CTCF bookmarked sites were specifically and prominently enriched. Cluster 5, which exhibits the fastest reactivation dynamics, was found to be particularly enriched in CTCF bookmarked sites compared to regions losing CTCF binding in mitosis. We conclude, therefore, that CTCF binding is associated with faster reactivation dynamics after mitosis, with CTCF bookmarking displaying the most robust association with gene reactivation. We next explored repli-ATAC data generated in ES cells (*Stewart-Morgan et al., 2019*), in an identical manner. We focused on accessibility measurements around Transcription Start Sites (TSSs) of ES cell expressed genes, which reflect binding of the pre-initiation complex of transcription. We found four clusters displaying different reactivation dynamics (Clusters 1 to 4; *Figure 7C*) and a further cluster showing minor changes in TSS accessibility after replication (Cluster 5; *Figure 7C*). Notably, only Cluster 4, which displays rapid reacquisition of accessibility, and Cluster 5, where TSSs remain accessible immediately after replication, show robust enrichments (*Figure 7D*). Therefore, both after replication and mitosis, the presence of CTCF binding sites correlates with the speed and efficiency of gene reactivation.

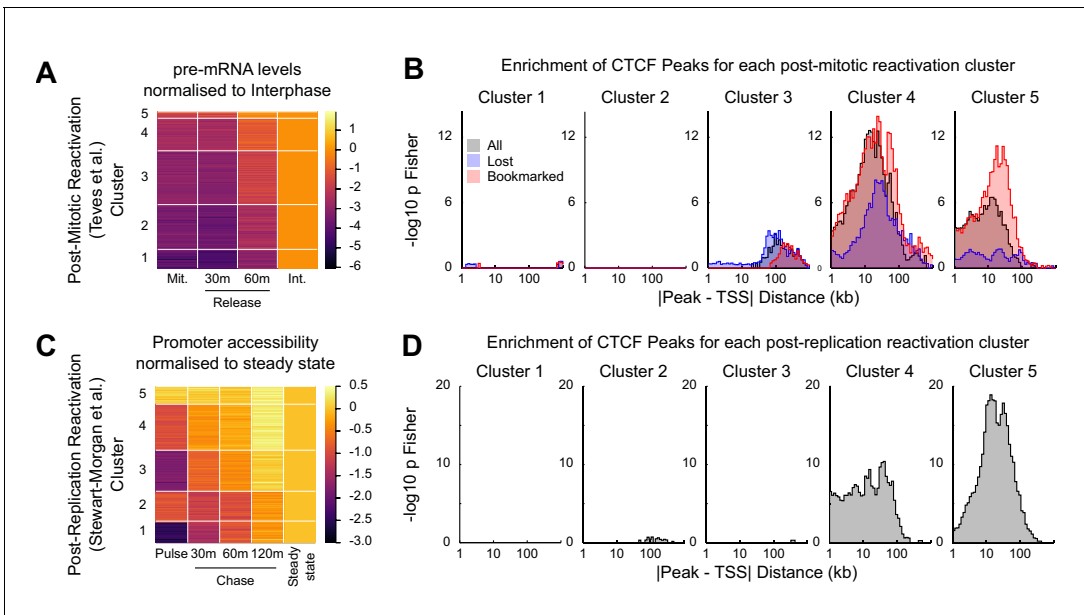

**Figure 7.** CTCF binding is associated with rapid gene activation after replication and mitosis. (A) K-means clustering (k = 5) of chromatin associated RNA-seq following release from mitosis (*Teves et al., 2018*). Heatmap shows $\log_2$ fold change of Mitosis (Mit.), 30 min and 60 min nocodazole release (30 m and 60 m respectively) relative to Interphase (Int.). (B) $-\log_{10}$ Fisher exact *p*-value enrichment of CTCF bookmarked and lost peaks in proximity to genes in clusters in (A). (C) K-means clustering (k = 5) of repli-ATAC-seq (*Stewart-Morgan et al., 2019*) at active promoters in mouse ES cells. Heatmap shows log2 fold change of Nascent (Pulse), 30 min (30 m), 1 hr (60 m) and 2 hr (120 m) chromatin relative to Steady State chromatin. (D) $-\log_{10}$ Fisher exact *p*-value enrichment of CTCF peaks in proximity to genes in clusters in (C).

DOI: https://doi.org/10.7554/eLife.47898.018

## Discussion

Here, we have studied the functional relationships between TF binding and nucleosome positioning during replication and mitosis, using CTCF as a paradigmatic example. While the role of CTCF in nucleosome positioning has already been suggested using siRNA knock-down (*Wiechens et al., 2016*), our work provides definitive evidence to conclude that CTCF is continuously required to maintain NDRs/NOAs at its binding sites. First, we describe a predictive relationship between the number and quality of CTCF binding motifs, the level of CTCF occupancy, and the degree of nucleosome positioning. Second, we observe a loss of nucleosome order 2 hr only upon CTCF depletion using the auxin degradation system. Given previous results, it is likely that CTCF positions nucleosomes by acting as a barrier to the nucleosome sliding activity of the chromatin remodeler SNF2H (*Barisic et al., 2019*; *Wiechens et al., 2016*). Furthermore, we show that the site-specific loss of CTCF binding in mitosis, which occurs at regions with poor binding motifs and reduced/absent Cohesin recruitment in interphase, is associated with a comprehensive loss of nucleosome order. Conversely, at regions preserving CTCF binding in mitosis, proper NDRs/NOAs are maintained; an observation which is underlined by the severely hypomorphic CTCF-aid which both lacks binding of CTCF and nucleosomal order at CTCF sites in mitosis. Hence, our data in interphasic and mitotic ES cells establish that CTCF plays a key role in maintaining local nucleosome order. In addition, our observations following DNA replication further indicate that CTCF not only maintains NDRs/NOAs, but also contributes to their establishment, as suggested by the rapid reinstatement of nucleosome positioning after the passage of the replication fork. This is independently supported by our analyses of post-replication chromatin accessibility (*Stewart-Morgan et al., 2019*) in ES cells and, notably, by previous observations linking CTCF binding to reduced DNA polymerase processivity and increased mutation rates at CTCF sites (*Smith and Whitehouse, 2012*; *Reijns et al., 2015*). We conclude, therefore, that CTCF is a major determinant of local nucleosome order, which orchestrates the organization of NDRs/NOAs throughout the cell-cycle. The direct and near instant dependence of nucleosome positioning on CTCF binding that we have described here, establishes the molecular basis for CTCF to build local chromatin resiliency.

Together with previous work (*Festuccia et al., 2019*), it appears that both CTCF and Esrrb, two zinc finger TFs, confer similar properties to their respective binding sites during replication and mitosis. Whilst CTCF comprises 10 C2H2-type and 1 C2HC-type zinc fingers (*Klenova et al., 1993*), Esrrb contains two C4-type zinc fingers – as is frequently observed by similar nuclear receptors (*Gearhart et al., 2003*). Hence, the number and type of zinc fingers do not seem to represent the intrinsic feature underlying the post-replication and mitotic activity of CTCF and Esrrb. It nevertheless remains possible that zinc finger TFs, which rapidly scan DNA for their motifs to engage in stable binding (*Iwahara and Levy, 2013*), are particularly suited to maintain nucleosome positioning after replication and during mitosis. Other C2H2 zinc finger TFs have been suggested to organize nucleosomes: in *Drosophila*, Zelda (*McDaniel et al., 2019*) and Phaser (*Baldi et al., 2018*); in mammals, REST and YY1 (*Valouev et al., 2011*; *Wang et al., 2012*). Whether these factors also organize nucleosomes during replication and mitosis requires further study; however, current data suggests that Zelda may act following replication (*McDaniel et al., 2019*) but not during mitosis (*Dufourt et al., 2018*). Properties in addition to their intrinsic DNA binding domains may thus be important for TFs to mediate their function during replication and mitosis. Notably, C2H2 zinc-fingers are typically phosphorylated during mitosis, preventing their site-specific interactions (*Dovat, 2002*; *Rizkallah et al., 2011*). Given our results, it is possible that either CTCF specifically, or C2H2 zinc fingers more generally, are not mitotically inactivated in ES cells. Moreover, not all mitotic bookmarking TFs (*Festuccia et al., 2017a*) are zinc finger proteins, as illustrated by FoxA1 (*Caravaca et al., 2013*) or Tbp (*Teves et al., 2018*). It will therefore be important to comprehensively identify TFs whose genomic targets exhibit nucleosome resiliency after replication and during mitosis. This will not only identify rules to explain the replication and mitotic function of some TFs, but will also address whether every mitotic bookmarking TF positions nucleosomes in mitosis and after replication.

Several studies have thus far assessed the mitotic bookmarking capacity of CTCF, with varying results (*Shen et al., 2015*; *Sekiya et al., 2017*; *Oomen et al., 2019*; *Burke et al., 2005*). Here, we show that CTCF binding in mitosis is cell type-specific, possibly reconciling these conflicting observations. Moreover, this indicates that the mitotic bookmarking activity of CTCF is a developmentally

regulated phenomenon, which may be mediated by differential mitotic phosphorylation, as discussed above. In support of developmental regulation, recent results have shown CTCF is capable of mitotic bookmarking in an erythroblast cell line (*Zhang, 2019*). Confirming to which cell types CTCF bookmarking is specific to, when during development it loses this activity, and how this impacts long-range chromatin interactions in mitosis and early in the following interphase, represent clear lines of investigation for the future. Nevertheless, since Cohesin is lost during prophase (*Nishiyama et al., 2013*), and is required to assemble Topological Associating Domains, CTCF bookmarking may not lead to the mitotic maintenance of these structures, as recently shown (*Zhang, 2019*). Similarly, it remains to be addressed whether the post-replication reestablishment of nucleosome order observed at CTCF binding sites is ES cell-specific. Indeed, we show here that nucleosomes are reorganized rapidly post-replication in ES cells at regulatory elements as compared to other cell types (*Ramachandran and Henikoff, 2016*). Nevertheless, our analysis indicates that different TFs exhibit drastic differences in their ability to reposition nucleosomes, with CTCF and Esrrb exhibiting a particularly compelling capacity to restructure nucleosomal arrays within minutes of the passage of the replication fork. Since ES cell self-renewal requires constant proliferation, the mechanisms that we have identified may contribute to the continuous maintenance and regulation of gene activity during successive cell divisions. The globally fast reorganization of the nucleosomes after replication, together with the particular capacity of CTCF and Esrrb to govern nucleosome positioning after replication and during mitosis, may indeed facilitate the reassembly of regulatory complexes in nascent chromatin fibers and in daughter ES cells (*Mirny, 2010*). While this hypothesis needs to be directly addressed, the correlation between post-replication/mitosis gene reactivation speed and CTCF binding, provides clear support. Therefore, a comprehensive investigation of TFs building local nucleosome resiliency throughout the cell-cycle, in ES cells and more generally during differentiation and development, will identify their role in the preservation of cell identity in proliferative cells. This will ultimately illuminate whether their activity bypasses the requirement for a robust epigenetic memory of active regulatory elements (*Reinberg and Vales, 2018*), particularly in cell types with increased plasticity such as ES cells (*Festuccia et al., 2017b*; *Festuccia et al., 2017a*).

## Materials and methods

### Key resources table

| Reagent type (species) or resource | Designation | Source or reference | Identifiers | Additional information |
|---|---|---|---|---|
| Cell line (*Mus musculus*) | E14Tg2a | | | Grown in serum and LIF |
| Cell line (*Mus musculus*) | Ctcf-aid | PMID: 28525758 | | Grown in serum and LIF |
| Antibody | anti-CTCF rabbit polyclonal | Active Motif | Cat# 61311 | 4 μL/ChIP-seq |
| Antibody | anti-SMC1 rabbit polyclonal | Bethyl Laboratories | Cat# A300-055A | 4 μL/ChIP-seq |
| Antibody | anti-H3 rabbit polyclonal | Abcam | Cat#: ab1791 | 5 μg/ChIP-seq |
| Software, algorithm | Bowtie2 v2.1.0 | (*Langmead and Salzberg, 2012*) | RRID: SCR_005476 | |
| Software, algorithm | STAR v2.5.0a | (*Dobin et al., 2013*) | RRID: SCR_015899 | |
| Software, algorithm | RSEM v1.2.29 | (*Li and Dewey, 2011*) | RRID: SCR_013027 | |
| Software, algorithm | VERSE | (*Zhu, 2016*) | https://github.com/qinzhu/VERSE | |

*Continued on next page*

*Continued*

| Reagent type (species) or resource | Designation | Source or reference | Identifiers | Additional information |
|---|---|---|---|---|
| Software, algorithm | MACS2 v2.0.10 | (*Feng et al., 2012*) | RRID: SCR_013291 | |
| Software, algorithm | DESeq2 v1.18.1 | (*Love et al., 2014*) | RRID: SCR_015687 | |
| Software, algorithm | FIMO v4.12.0 | (*Grant et al., 2011*) | RRID: SCR_001783 | |
| Software, algorithm | picard-tools v2.8.1 | http://broadinstitute.github.io/picard/ | RRID: SCR_006525 | |
| Software, algorithm | samtools v1.9 | (*Li et al., 2009*) | RRID: SCR_002105 | |
| Software, algorithm | GPy | https://github.com/SheffieldML/GPy | | |
| Software, algorithm | Julia v0.6.4 | (*Bezanson et al., 2017*) | https://julialang.org/ | |
| Software, algorithm | Python v3.6.7 | https://www.python.org/ | RRID: SCR_008394 | |
| Software, algorithm | R v 3.4.4 | http://www.r-project.org/ | RRID: SCR_001905 | |

## Cell culture and mitotic preparations

ES cells (wild-type E14Tg2a and CTCF-aid derivatives) were cultured on 0.1% gelatin (SIGMA, G1890-100G) in DMEM+GlutaMax-I (Gibco, 31966–021), 10% FCS (Gibco 10270–098), 100 µM β-mercaptoethanol (Gibco, 31350–010), 1 × MEM non-essential amino acids (Gibco, 1140–035) and 10 ng/ml recombinant LIF (MILTENYI BIOTEC, 130-099-895). ES cells were passaged 1:10 every 2–3 days. NIH3T3 and C2C12 cells were cultured in DMEM + GlutaMax-I supplemented with 10% FCS. NIH3T3 and C2C12 were passaged 1:15 and 1/20, respectively, every 3 days. To obtain mitotic ES cells (>95% purity as assessed by DAPI staining and microscopy), we used a nocodazole shake-off approach, as described before (*Festuccia et al., 2019*). To deplete CTCF-aid in interphase, cells were treated with 0.5 mM auxin (IAA Sigma, I5148) for 2 hr; to deplete CTCF-aid in mitosis, the cells were first treated with nocodazole for 4 hr and then with nocodazole and IAA for 2 hr, after which they were harvested by shake-off. C2C12 cells were synchronized like ES cells except that slightly longer nocodazole treatment (7 hr) was required to improve the yield. NIH3T3 cells were synchronized with a triple approach. First, $1.5 \times 10^6$ cells were seeded in 150 mm dishes and grown with 2 mM thymidine for 18 hr to enrich in cells arrested at the G1/S transition. After two washes with PBS, they were released for 8 hr in regular medium before being arrested again for 17 hr with 2 mM Thymidine. After the double thymidine block the cells were washed twice with PBS and released in regular medium for 7 hr. Finally, cells were incubated with nocodazole for 6 hr and mitotic cells isolated by gentle tapping of the dishes against an immobilized surface.

## MINCE-seq

The protocol was developed based on three existing protocols (*Ramachandran and Henikoff, 2016*; *Kliszczak et al., 2011*; *Sirbu et al., 2011*); for the CLICK reaction we used radical-free aerobic conditions, as previously described (*Presolski et al., 2011*).

Pulse/Chase with EdU: $15 \times 10^6$ live cells were seeded per 150 mm dish 24 hr prior to the beginning of each experiment (three dishes per condition). The cells were then incubated with pre-warmed medium containing 5 µM EdU for 2.5 min at 37°C. Next, the cells were either harvested and fixed (pulse) or, in parallel, washed with pre-warmed medium and incubated for 1 hr in the presence of 100 µM Thymidine for 1 hr at 37°C (chase).

These samples and the respective controls (*Figure 4—figure supplement 1*) were checked by FACS after fixation (4% para-formaldehyde, 15 min at room temperature protected from light), permeabilization (PBS 1% BSA/0.1% Saponin for 15 min at room temperature), incubation with 100 µl

CLICK reaction Mix (see below), and labeling with Streptavidin-Alexa Fluor 488 (S32354, Thermo Fisher; diluted 1:1300 in permeabilization solution) for 1 hr at room temperature in the dark.

Micrococcal Nuclease digestion (MNase): after trypsinization, $120 \times 10^6$ cells per sample were fixed with 1% formaldehyde (Thermo, 28908; 10 min at room temperature with occasional mixing) and quenched with 0.125M Glycine (5 min). Fixed cells, after being washed twice with PBS, were lysed for 10 min on a rotating wheel at 4°C in 5 mL of lysis buffer (50 mM HEPES pH 7.8/150 mM NaCl/0.5% v/v IGEPAL/0.25% v/v Triton-X/10% v/v Glycerol) supplemented with $1 \times$ protease inhibitor cocktail (PIC-Roche, 04 693 116 001). After centrifugation, the cells were resuspended in 4 ml Wash Buffer (WB: 10 mM Tris-HCl pH 8/200 mM NaCl) supplemented with 0.5 mM DTT and incubated for 10 min on a rotating wheel at 4°C. Nuclei were pelleted for 5 min at 3000 rpm at 4°C and resuspended in 1.5 ml RIPA buffer (10 mM Tris-HCl pH 8/140 mM NaCl/1% Triton-X/0.1% Sodium-Deoxycholate/0.1% SDS) freshly supplemented with $1 \times$ protease inhibitor cocktail and 1 mM $CaCl_2$. Samples were equally split in 3 (1.5 ml tubes) and incubated for 10 min in a water bath at 37°C. MNase (Thermo Scientific EN0181) was diluted at 80 U/μl in RIPA buffer freshly supplemented with $1 \times$ protease inhibitor cocktail and 1 mM $CaCl_2$. The digestion was carried out by adding MNase at a final concentration of 6U per $10^6$ cells and incubation for 5 min at 37°C with occasional mixing. Digestions were stopped by placing tubes on ice and by adding $2 \times$ STOP Buffer (2% Triton/0.2% SDS/300 mM NaCl/10 mM EDTA). Digested chromatin was recovered by overnight rotation at 4°C followed by centrifugation at 4°C at maximum speed for 10 min. The chromatin was equally distributed in 1.5 ml tubes and the crosslinking reversed overnight at 65°C in the presence of 1% SDS and shaking (800 rpm). The next day the samples were incubated for 2 hr at 56°C with 5 μl Proteinase K (GEXPRK01B5 Eurobio) and processed through phenol/chlorophorm extraction and ethanol precipitation. DNA was resuspended in a total volume of 0.4 ml UltraPure Nuclease free water. RNase digestion was carried out with 5 μl RNase A (EN0531 Thermo Scientific) for 2 hr at 37°C. The DNA was purified and precipitated again and resuspended in 51 μl UltraPure Nuclease free water. 1 μl of DNA was further diluted and the fragment sizes were analyzed with a D1000 HS Screentape (Agilent 5067–5584) on an Agilent Tapestation.

CLICK reaction and Streptavidin Pull-Down: each DNA sample (50 μl) was supplemented, in order, with the following buffers: 342.5 μl of 100 mM Potassium Phosphate Buffer (15433339 Fisher Scientific); 50 μl of 10 mM Biotin-TEG Azide (762024 Sigma) resuspended in DMSO; 7.5 μl of pre-mixed Cupric Sulfate Pentahydrate (2.5 μl at 20 mM; C8027 Sigma) and THTPA (5 μl at 50 mM; 762342 Sigma); 25 μl of 100 mM Aminoguanidine Hydrochloride (396494 Sigma); 25 μl of 100 mM Sodium Ascorbate (A4034 Sigma). After briefly vortexing, the reactions were performed for 1.5 hr at room temperature protected from light, and DNA was subsequently ethanol-precipitated and resuspended in 400 μl UltraPure water. 20 μl were kept as input and the rest (380 μl) were used for streptavidin pull down. For this, 25 μl M-280 streptavidin magnetic beads (10292593 Thermo Scientific) per reaction were washed three times (10 min rotation at room temperature per wash) with 1 ml $1 \times$ Wash and Binding buffer (W and B); ($2 \times$ W and B: 10 mM Tris-HCl pH 7.5; 1 mM EDTA; 2M NaCl; 0.2% Tween 20) and resuspended in 25 μl $1 \times$ W and B buffer. EdU-labeled DNA was then pulled down with 380 μl $2 \times$ W and B buffer and 25 μl washed beads (1 hr at room temperature on a rotating wheel). Beads were separated by placing the tubes on a magnetic rack and the unbound fraction was stored at −20°C. The beads were washed 3 times with 1 ml W and B buffer (10 min at room temperature on a rotating wheel). Washed beads were resuspended in 100 μl TE/1% SDS and DNA was eluted with 20 μl Proteinase K (1 hr at 65°C with occasional vortexing). After adding TE to a final volume of 200 μl, the DNA was phenol/chloroform-extracted, ethanol-precipitated and resuspended in 20 μl UltraPure water.

Library preparation and sequencing: we used 10 ng of DNA from the input samples and the whole pulled-down samples to prepare libraries as previously described (*Festuccia et al., 2019*). Sequencing (PE150) was performed by Novogene Co. Ltd. Only samples from experiments with non-detectable levels of DNA in control samples where biotin was replaced by DMSO during the CLICK reaction were used for library preparation.

## ChIP-seq

For interphase, $2.5 \times 10^6$ fixed cells (1% formaldehyde for 10 min followed by 5 min with 0.125 M Glycine) were resuspended in 2 ml swelling buffer (25 mM Hepes pH 7.95, 10 mM KCl, 10 mM EDTA; freshly supplemented with $1 \times$ protease inhibitor cocktail – 04 693 116 001 PIC-Roche, and

0.5% IGEPAL), incubated for 30 min on ice, and passed 40 times in a dounce homogenizer to recover nuclei. For mitotic cells, the homogenization step was omitted. Mitotic and interphase cells were then treated in parallel to sonicate the chromatin in 300 µl of TSE150 (0.1% SDS, 1% Triton X-100, 2 mM EDTA, 20 mM Tris-HCl pH8, 150 mM NaCl; freshly supplemented with 1 × PIC) using 1.5 ml tubes (Diagenode) and a Bioruptor Pico (Diagenode; seven cycles divided into 30 s ON–30 s OFF sub-cycles at maximum power, in circulating ice-cold water). After centrifugation (30 min, full speed, 4°C), the supernatant was pre-cleared for 2 hr with 50 µl of protein G Sepharose beads (P3296-5 ML Sigma) 50% slurry, previously blocked with BSA (500 µg/ml; 5931665103 Roche) and yeast tRNA (1 µg/ml; AM7119 Invitrogen). 20 µl were set apart (input) before over-night immunopre-cipitation at 4°C on a rotating wheel with 4 µl of anti-CTCF (Active Motif 61311) or 4 µl of anti-SMC1 (Bethyl Laboratories A300-055A) antibodies in 500 µl of TSE150. Protein G beads (25 µL 50% slurry) were added for 4 hr rotating on-wheel at 4°C. Beads were pelleted and washed for 15 min rotating on-wheel at 4°C with 1 ml of the following buffers: three washes with TSE150, one wash with TSE500 (as TSE150 but 500 mM NaCl), one wash with Washing buffer (10 mM Tris-HCl pH8, 0.25M LiCl, 0.5% NP-40, 0.5% Na-deoxycholate, 1 mM EDTA), and two washes with TE (10 mM Tris-HCl pH8, 1 mM EDTA). Elution was performed in 100 µl of elution buffer (1% SDS, 10 mM EDTA, 50 mM Tris-HCl pH 8) for 15 min at 65°C after vigorous vortexing. Eluates were collected after centrifugation and beads rinsed in 150 µl of TE-SDS1%. After centrifugation, the supernatant was pooled with the corresponding first eluate. For both immunoprecipitated and input chromatin, the crosslinking was reversed overnight at 65°C, followed by proteinase K treatment, phenol/chloroform extraction and ethanol precipitation.

H3 ChIP-seq was performed on MNase-digested chromatin using $2.5 \times 10^6$ cells fixed with form-aldehyde as described above. Fixed cells were resuspended in 500 µl of MNase buffer (50 mM Tris-HCl pH8, 1 mM CaCl2, 0.2% Triton X-100) supplemented with protease inhibitor cocktail (PIC; 04 693 116 001 Roche). Cells were pre-incubated for 10 min at 37°C with 16U of MNase (Expressed in Kunitz, 1 Kunitz is equivalent to 10 gel units, NEB M0247S) added to the reaction. Cells were incubated for further 10 min at 37°C, inverting the tubes occasionally. The reaction was stopped on ice by adding 500 µl of 2 × STOP buffer (2% Triton X-100, 0.2% SDS, 300 mM NaCl, 10 mM EDTA). Tubes were left rotating overnight on a wheel to allow diffusion of the digested fragments. The cell suspension was spun down and the supernatant stored at −80°C. Subsequently, 50 µl of chromatin were brought to a final volume of 500 µl with TSE150 (0.1% SDS, 1% Triton X-100, 2 mM EDTA, 20 mM Tris-HCl pH8, 150 mM NaCl), freshly supplemented with protease inhibitor cocktail (PIC-Roche, 04 693 116 001), and ChIP was performed as described above using 5 µg of anti-Histone H3 rabbit polyclonal (Abcam; ab1791).

All libraries were generated as previously described (*Festuccia et al., 2019*) and sequenced in the BioMics facility of the Institut Pasteur.

## Western blot

For Western Blot analysis cell pellets corresponding to $10^6$ cells were resuspended in 100 µl Laemmli Sample Buffer (161–0737 BIO-RAD) containing β-mercaptoethanol. Lysed pellets were boiled for 10 min at 95°C and centrifuged for 10 min at maximum speed at room temperature. Typically, 10 µl per sample were loaded on 4–15% Mini-PROTEAN TGX Stain-Free Gels (4568086 BIO-RAD) and run in 1 × SDS Running Buffer (250 mM Tris/1.92M Glycine/1% SDS) at 10-20mA using the Mini-PROTEAN Tetra System (BIO-RAD). Proteins were transferred on nitrocellulose membranes (Amersham Protran 10600003) for 1 hr (or 3 hr for CTCF) at 300mA using the wet transfer system (BIO-RAD) in 1 × Transfer Buffer (10 × 0.25M Tris/1.92M Glycine) prepared with a final concentration of 20% Ethanol. Membranes were blocked in PBST (PBS 0.1% Tween-20) 5% BSA for 1 hr at room temperature and incubated overnight at 4°C with primary antibodies diluted in PBST 5% BSA (1:5000 anti-H3 Abcam ab1791; 1:500 anti-CTCF Active Motif 61311). Excess antibodies were washed off with PBST (five washes, 5 min each) and incubated for 1 hr at room temperature in secondary antibodies HRP-conjugated diluted in PBST 5% BSA (1:10000 anti-Rabbit IgG-HRP Thermo Fisher RB230254). Membranes were washed five times, 10 min each at room temperature and incubated with PIERCE ECL2 Western Blotting Substrate (80196 Thermo Scientific) 5 min in dark. After excess reagent was removed, proteins were visualized using the BIO-RAD Chemidoc MP Imaging System and processed using the Image Lab Software (BIO-RAD).

## Chromatin fractionation

Ten million asynchronous or mitotic ES cells, prepared as described above, were washed in PBS1X and either fixed with Formaldehyde or directly resuspended in 200 µL of Buffer A (10 mM HEPES pH 7.9/10 mM KCl/1.5 mM MgCl2/0.34 M Sucrose/10% Glycerol/1 mM DTT), supplemented with complete protease inhibitors and 0.1% Triton X-100. Cells were incubated for 8 min on ice. After recovering the nuclei by centrifugation (1300 rpm; 4°C), the pellet was washed in Buffer A and then lysed with 100 µL of Buffer B (3 mM EDTA/0.2 mM EGTA/1 mM DTT) supplemented with complete protease inhibitors for 30 min on ice. After spinning down, the pellet (chromatin fraction) was washed in Buffer B and resuspended in 60 µL of 2x reducing Laemmli buffer supplemented with 5% Bmercaptoethanol). The samples were sonicated 10 cycles (30seg ON/30 seg off) in a Bioruptor Pico and boiled for 35 min at 95°C. After 10 min centrifugation at max speed, the samples were processed for CTCF and H3 western-blot as described above.

## Imaging

Immunostaining of ES, C2C12 and NIH3T3 cells: cells were plated at a density of $2 \times 10^4$ on IBIDI hitreat plates coated overnight with poly-L-ornithine (0.01%; Sigma, P4957) at 4°C, washed and coated 2 hr with laminin (10 µg/ml in PBS; Millipore, CC095). Two days post-seeding, cells were washed twice with PBS, fixed for 10 min at room temperature with PBS 4% Formaldehyde (Thermo, 28908), washed twice in PBS and permeabilized with PBS/0.1% v/v Triton X-100 supplemented with 3% of donkey serum (Sigma, D9663) for 15 min at room temperature. Incubation with primary anti-CTCF (1:500; Active Motif 61311) and anti-SMC1 (1:500; Bethyl Laboratories A300-055A) antibodies were performed in PBS with 3% donkey serum. After three washes in PBS/0.1% Triton X-100, Alexa Fluor 594 AffiniPure Donkey Anti-Rabbit secondary antibodies (Jackson ImmunoResearch, 711-585-152) were applied for 2 hr at room temperature (2 µg/ml). Cells were washed three times in PBS/0.1% v/v Triton X-100, nuclei counterstained with 4',6-diamidino-2- phenylindole (DAPI; Sigma, D9542), and imaged with a LSM800 Zeiss microscope using a 64 × or 40 × oil immersion objective.

### Live imaging

CTCF-aid ES cells also carry a GFP fused in frame with CTCF-Aid (*Nora et al., 2017*), enabling live imaging of CTCF. To do this, the cells were cultured on IBIDI plates in phenol red-free medium (Gibco, 31053–028) and incubated with 500 nM Hoechst-33342 for 20 min before imaging, which was performed at 37°C in a humidified atmosphere (7% CO2). Images were acquired with a 63 × oil immersion objective on a Zeiss AxioObserver Z1 microscope equipped with a Yokogawa CSUX1 spinning-disk confocal scanner, a Hamamatsu EMCCD ImageEM X2 camera using the Volocity acquisition software.

### Immunostaining of embryos

All experiments were conducted according to the French and European regulations on care and protection of laboratory animals (EC Directive 86/609, French Law 2001–486 issued on June 6, 2001) and were approved by the Institut Pasteur ethics committee (n° dha180008). CD1 (Charles River Laboratories, France) embryos were recovered at E3.5 (mid-blastocyst stage) and cultured for 6 hr in EmbryoMax KSOM (Millipore Bioscience, MR-121D) with or without 5 µM Nocodazole (Tocris Biotechne, 1228) to enrich for mitotic cells. Embryos were fixed for 15 min at room temperature in 4% paraformaldehyde (Euromedex, 15714), rinsed with 1 × PBS and incubated in 1 × PBS, 0.1% Triton-X100, 10% donkey serum for 30 min at RT. Embryos were then incubated with primary antibodies overnight at 4°C and with secondary antibodies for 1 hr at RT in 1 × PBS, 0.1% triton-X100, 10% donkey serum. The following antibodies were used: anti-Nanog (1:200; eBioscience 14-5761-80), anti-CTCF (1:300; Active Motif 61311), donkey anti-rat Alexa Fluor 488 (1:300; Invitrogen A21208) and donkey anti-rabbit Alexa Fluor 546 (1:300; Invitrogen A10040). Nuclei were counterstained with Hoechst 33342 (1.6µM; Sigma Aldrich 14533), and embryos were placed individually in glass bottom microwell dishes (MatTek Corporation P35G-1.5–20 C) in 1 × PBS. Fluorescent images were obtained using a confocal laser-scanning microscope (LSM800; Zeiss) with the objective Plan-apochromat 20×/NA 0.8, speed 6, pinhole one airy unit, and laser intensities suited for optical section thickness of 2 µm. Images were analyzed and processed using Fiji and Photoshop CS6 softwares. Pictures correspond to a projection of 5 confocal optical slices.

## Computational methods
### Data and availability
Samples are summarized in *Supplementary file 1*. Briefly for ChIP-seq, we sequenced 3 x Interphase and 3 x Mitosis wild-type CTCF; 2 x Interphase and 2 x Mitosis wild-type SMC1; 2 x Interphase and 2 x Mitosis CTCF in CTCF-aid for -IAA and +IAA (2 hr); 1 x interphase and 1 x mitosis of CTCF in C2C12 and NIH3T3 cells. For MINCE-seq we sequenced 2 x input and IP pairs for pulse and chase samples. Finally, MNase H3 ChIP-seq in CTCF-aid 2 x Interphase and 2 x mitosis for both -IAA and +IAA (2 hr). We previously generated wild-type MNase-seq: 3 x Interphase and 3 x Mitosis; MNase H3 ChIP-seq: 2 x Interphase and 2 x Mitosis; and ATAC-seq: 2 x Interphase and 2 x Mitosis. All previously generated datasets are described (*Festuccia et al., 2019*) and are available from GEO accession: GSE122589.

## CTCF and SMC1 binding in interphase and in mitosis
### Data Processing
Reads were aligned with Bowtie 2 (*Langmead and Salzberg, 2012*) to the mm10 genome, with options '-k 10'. Reads were additionally filtered for those with a single discovered alignment (in Bowtie 2 'k' mode this is mapping quality = 255) and an edit distance less than 4. All libraries were constructed with custom unique molecular identifier barcode as described before (*Festuccia et al., 2019*). Therefore, reads aligning with identical position, strand and barcode were treated as duplicates and collapsed.

### Peak calling
We used a strategy previously described (*Festuccia et al., 2019*), where peaks were called against relevant inputs/controls for all samples using MACS2 (*Feng et al., 2012*) with 'callpeak -q 0.2 g mm'. Peaks intersecting with the mm10 blacklist (*ENCODE Project Consortium, 2012*) were excluded along with those on chrM and chrY. To determine a set of candidate binding regions for each factor we merged all peaks in interphase and mitosis. Peaks were further filtered to have MACS2 FDR < 0.01 in both replicates in either interphase or mitosis, and a height >0.8 reads per million (RPM) in at least one sample (a height of ~16 raw reads at our mean mapped read depth of ~20 million reads).

### Bookmarking analysis
We used a strategy previously described (*Festuccia et al., 2019*), where we take an approach similar to differential expression analysis in RNA-seq to determine the set of peaks bookmarked CTCF and SMC1. We combine the read counts for input and ChIP samples for a given TF in a single generalized linear model (GLM) aimed at assessing the difference between input and ChIP signal in both interphase and mitosis at each peak. We implement this GLM using DESeq2 (*Love et al., 2014*). We expect that nearly all peaks will have significant differences between input and ChIP, and that the majority of peaks will have significantly different occupancy between interphase and mitosis. Therefore, we set the size factors to the total mapped reads of each sample. We encode ChIP versus Input by a factor ChipTF and mitosis samples by a factor ChipM, and then we ran Wald tests on the model ~ChipTF + ChipM. At significance FDR < 0.05 and without independent filtering, we tested ChipTF (the fold change of interphase over input), the sum ChipTF + ChipM (the fold change of mitosis over input), and finally ChipM the difference between interphase and mitosis. To determine the set of bookmarked peaks we required that a peak had differential occupancy (ChipTF + ChipM, FDR < 0.05), that both mitosis replicates had MACS2 FDR < 0.01 with one mitosis replicate had MACS2 FDR < $10^{-10}$. Combined, this is a conservative strategy that helps ameliorate the effects of contamination from interphase cells (*Festuccia et al., 2019*). The significance of ChipM was further used to classify three types of bookmarking peaks with either higher enrichment in interphase (BI), in mitosis (BM), or similar in both phases (BS). This resulted in:

CTCF: Lost (L): 22,550, BI: 18,723, BS: 10,456, BM: 360
SMC1: (L): 17,251

## Visualization and normalization of ChIP-seq data

Single end read data was extended to mean fragment length (~200 bp for CTCF-aid samples and ~120 bp otherwise). For CTCF, the single base pair at the center of the inferred fragment was marked and normalized to units of reads per million (RPM), with the exception of CTCF-aid mitosis, interphase and mitosis comparison between E14Tg2a (ES cells), C2C12 and NIH3T3 cells, due to low signal in some of these samples. In these cases, the entire fragment was marked and normalized to be comparable to single base heatmaps. SMC1 heatmaps were generated by marking entire fragments. All heatmaps are ordered by descending peak height and are at resolution of 100 sites per pixel. Heatmaps color scale is linear (inferno colormap) scaled to 0.2 of wild-type interphase (or mitosis for *Figure 6—figure supplement 1A*) peak maximum.

## Transcription Factor Motif Detection

To discover CTCF binding motifs we scanned the mm10 genome using FIMO (*Grant et al., 2011*) for motif MA0139.1 with parameters '–thresh 1e-3 –max-stored-scores 50000000' and supplied a 0-order Markov background file describing the relative nucleotide frequencies in the mm10 assembly. Motifs were intersected with peaks. Metaplots, V-plots and heatmaps are centered on the best scoring motif within each peak.

## Nucleosome positioning MNase-seq, MNase-H3-ChIP-seq, MINCE-seq

### Data Processing

Paired end reads were trimmed by aligning read pairs to discover regions of reverse complementarity surrounded by our custom adapters. Alignment and trimming were performed with the BioSequences package for Julia 0.6 (*Bezanson et al., 2017*). Reads were aligned to mm10 genome using Bowtie 2 (*Langmead and Salzberg, 2012*) with options '-k 10 -I 0 -X 1000 –no-discordant –no-mixed', and filtered for reads with a single alignment mean edit distance less than four between read pairs. For all libraries with barcoded adapters, identical barcodes with identical position were collapsed.

### Bias correction

In all MNase samples we estimated and corrected the MNase cutting bias. We use our previously applied approach (*Festuccia et al., 2019*), which is a simplified version of seqOutBias (*Martins et al., 2018*). We evaluated the relative rates of occurrence of the k-mer at the end of each read and the total occurrence of that k-mer in the genome. Specifically, we took the two 6-mers lying over the end coordinates of each read, such that each 6-mer was composed of two 3-mers, one lying within the read and one lying outside. We counted total k-mers in the genome using the BioSequences package in Julia 0.6 (*Bezanson et al., 2017*). To assess an appropriate correction, we averaged the k-mers at the end of each read in a position weight matrix (PWM), we found the left PWM to be approximately equal to the reverse complement of the right PWM, and so we calculated a correction based on the left k-mer and the reverse complement of the right k-mer. If $\gamma_i$ is the rate of occurrence of k-mer $k_i$ in the genome, $\rho_i$ is the rate of occurrence of the same k-mer at MNase cutsites, and $k_L$ is the left k-mer and $k_{R^\dagger}$ is the reverse complement of the right k-mer, then in all analysis each fragment is weighted by: $\sqrt{\rho_L \rho_{R^\dagger} / \gamma_L \gamma_{R^\dagger}}$. We employed a pseudocount of 100 in calculating k-mer rates.

### MNase fragment size selection

We employed V-plots (*Figures 1B*, *3C*, *5A* and *6B*) over features to conservatively and robustly select fragment windows for footprinting and evaluating nucleosome signal. For MNase samples, we selected fragments less than 100 bp for footprints and fragments in the interval [140, 200]bp for nucleosomes. For MINCE-seq libraries which exhibited a different fragment size distribution we used fragments in the interval [120, 200] bp for nucleosomes.

### Nucleosome normalization

For wild-type analysis we calculated the rate of mid-points of nucleosomal size fragments per billion (mid-points per billion - MPB) at each base pair per site for heatmaps and single loci, and averaged

over all sites for metaplots and V-plots. For MINCE-seq and CTCF-aid MNase H3 ChIP-seq which exhibited more variation in fragment size distributions we calculated the MPB relative to total nucleosomal size fragments. To focus on differences of nucleosomal organization in MINCE-seq, we normalized total nucleosomal fragments within [−1 kb, +1 kb] of the center of each feature in pulse and chase to be equal to the relevant input. Similarly, for CTCF-aid samples we normalized total nucleosomes per feature in +IAA to be equal to -IAA, and in (*Figure 6C,E*) we further normalized to total wild-type interphase nucleosomes.

## Nucleosome visualization
All heatmaps mark the midpoint of nucleosomal sizes MNase or MNase-H3 nucleosomal fragments in MPB, displayed at 100-site per pixel resolution with inferno colormap scaled to wild-type interphase MNase or MNase H3 maximum as appropriate. Metaplots are calculated at base pair resolution by the mean MPB per base pair per site surrounding each feature (visualized as the point clouds in *Figures 3D*, *4A* and *6C*); we then apply Gaussian process regression to smooth and model nucleosome positions as described below.

## Nucleosome positioning regression
To assess the statistics of the mean nucleosome signal over a set of regions we employed Gaussian process regression (*Rasmussen and Williams, 2006*) on MNase and MNase H3 metaplots. We used a squared exponential covariance function and selected hyper-parameters for signal variance, length scale and noise variance, optimized on [-500, 500] bp interval surrounding the central point of each feature. For all but CTCF-aid data, this contains the signal of primary significance and importantly the covariances are relatively stationary over this region as compared to outside where length and noise scales change as the data loses coherence. For CTCF-aid +IAA interphase and +/- IAA mitosis data the nucleosomes roll inwards over the region of maximum MNase-cutting bias (visible as V in V-plot *Figure 3C*), causing an artificial break in the +/-1 nucleosomes with altered covariance. Therefore, we optimized hyperparameters on the union [-500, -100] U [100, 500] bp. We then use these optimized hyperparameters to predict over the full region. We selected hyperparameters by sparse Gaussian process regression employing a variational approximation to the marginal likelihood (as offered by https://github.com/SheffieldML/GPy) (*Titsias, 2009*), with an initial inducing inputs every 10bp. To account for the overdispersed count-data nature of the nucleosome fragment mid-points, we employ a log transform to stabilize variance and then assume a Gaussian likelihood for Gaussian process regression. We transform the total midpoints $m_i$ at site $i$ data by: $y_i = \log(\alpha m_i + \beta)$, we select $\alpha = 10^3$ and $\beta = 1$. We make predictions and invert the transform, $\hat{m}_i = (e^{\hat{y}_i} - \beta)/\alpha$ where $\hat{y}_i$ is the Gaussian process mean, thus $\hat{m}_i$ reflects the median of the transformed random variable and we plot this median.

## Nucleosome positioning
To assess the position of the +/- 1 nucleosomes and their dependence on CTCF occupancy, we took the nucleosome fragment midpoints in [-230, -70] bp for the -1 nucleosome and [70, 230] bp relative to CTCF motif center in 100 site bins descending with CTCF peak size. For each bin we evaluated the median of the empirical cumulative density function within each region. If $m_{ij}$ is the total number of nucleosome fragment midpoints at base $i$ in site $j$, we calculate $\hat{F}_j(i) = \left(\sum_{k \leq i} m_{kj}\right)/\left(\sum_k m_{kj}\right)$, and take the median as the smallest $i$ such that $\hat{F}_j(i) \geq 0.5$. We then smooth the medians per 100-site bins by Gaussian process regression. To asses nucleosome positioning over a NOA averaged over a set of sites we call the maxima of the Gaussian process median described above.

## Nucleosome spectral density
To assess nucleosome periodicity we report the spectral density (*Figure 4B*) of the covariance function with optimized hyperparameters. For the squared exponential kernel, frequency $s$ and length-scale $l$ this is given by *Rasmussen and Williams (2006)*: $S(s,l) = \sqrt{2\pi l^2} \exp(2\pi^2 s^2 l^2)$, we evaluate spectral density at period p = 180, s = 1/180 which corresponds to nucleosome plus linker.

## Chromatin accessibility in interphase and in mitosis

Paired end reads were trimmed by aligning read pairs to discover regions of reverse complementarity surrounded by Nextera sequencing adapters for ATAC-seq. Alignment and trimming was performed with the BioSequences package for Julia 0.6 (*Bezanson et al., 2017*). Reads were aligned to mm10 genome using Bowtie 2 (*Langmead and Salzberg, 2012*) with options '-k 10 -l 0 -X 1000 –no-discordant –no-mixed', and filtered for reads with a single alignment mean edit distance less than four between read pairs. To generate heatmaps, the two end points (cut sites) of fragments in the 0–100 bp range, shifted inward by +/- 4 bp as recommended (*Buenrostro et al., 2013*) and piled at base pair resolution. Heatmaps are visualized at a resolution of 100 sites per pixel using inferno colormap scaled to 0.5 maximum interphase signal.

## Genomic features

All heatmaps are metaplots are centered on CTCF maximal motifs (*Supplementary file 2*); SMC1 summits (*Supplementary file 2*); Esrrb motifs (*Figure 4A*) for Esrrb bookmarked regions were Esrrb dictates nucleosomal organization as previously defined (*Festuccia et al., 2019*); Oct4/Sox2 composite motifs (*Figure 4A*) at Oct4 and Sox2 interphase binding regions as defined (*Festuccia et al., 2019*); P300 binding regions centered of summits derived from the Encode portal (*ENCODE Project Consortium, 2012*; *Davis et al., 2018*), experiment identifier: ENCSR000CCD, file: ENCFF179FJG. We further restricted P300 binding regions by ChromHMM ES enhancers (*Ernst and Kellis, 2012*) (https://github.com/guifengwei/ChromHMM_mESC_mm1) (*Pintacuda et al., 2017*). We retain p300 binding regions that intersected as regions annotated as 'Enhancer', 'Strong Enhancer' or 'Weak/Poised Enhancer' and no other ChromHMM categories. To asses CTCF binding at different chromatin states (*Figure 3D*) we used the same ChromHMM data and selected insulators, the enhancer set described above, active promoters and active gene bodies ('Transcription Elongation').

## Post-mitotic reactivation - Chromatin associated RNA-seq

Chromatin associated RNA-seq reads were taken from GSE109964 (*Teves et al., 2018*) and aligned to an index comprising the mm10 genome and the ERCC spikes, using STAR (*Dobin et al., 2013*) as part of the RSEM-STAR pipeline (*Li and Dewey, 2011*) with additional options '–seed 1618 –star-output-genome-bam –calc-pme –calc-ci –estimate-rspd'. To focus on pre-mRNA signal STAR genome bam files were further quantified to count intronic reads. First duplicates were removed using Picard (https://broadinstitute.github.io/picard/) MarkDuplicates with options "TAGGING_PO-LICY=All ASO=coordinate READ_NAME_REGEX=null". Reads arising unambiguously from spliced mRNAs or unspliced pre-mRNAs were counted using VERSE (*Zhu, 2016*), we generated a custom gtf file with additional with the feature field `transcript` entries marking the extent of each isoform in the Ensembl (*Zerbino et al., 2018*) 93 release with ERCC spikes added. VERSE was run in strand specific mode with options '-s 2 –ignoreDup –singleEnd –multithreadDecompress -T 12 -z 2 -t 'exon; transcript'. To normalize by ERCC spikes we took a strategy that reduces spike-in technical noise (*Owens et al., 2016*), we first normalized for sequencing depth by the total exonic and transcript reads per library to a mean depth of 15 million reads, we then averaged depth normalized RNA spikes between biological replicates and calculated and applied a correction factor per biological condition. If $s_{C_j}$ is the total depth normalized exonic counts of all spikes in condition $C$ sample $j$, we calculate the average spike per condition $\bar{s_C}$ and $\sigma_C = \left(\sum_c \bar{s_c}\right)/n\bar{s_C}$ where $n$ is the total number of conditions. We then correct the depth normalised transcript (intronic) count by the spikes, for gene $i$ in sample $C_j$ as $g_{iC_j} = \sigma_C g_{iC_j}$. Genes were filtered for those in which all replicates in at least two conditions had a spike-corrected depth normalized transcript count of greater than 10, and whose interphase mean spike corrected RPKM was greater than 2/15. RPKMs were calculated by the normalizing counts by the total number of non-exonic bases per transcript model. This resulted in 13,233 filtered genes. To determine the groups of genes with differing reactivation behavior we used k-means clustering as offered by the Clustering package of Julia (*Bezanson et al., 2017*). We normalized spike corrected mean counts of mitosis, 30 min release and 60 min release to interphase as $\log_2$ fold changes. We clustered with $2 \leq k \leq 20$, and compared cluster assignments for $k$ to $k + 1$ we found the Rand Index (*Rand, 1971*), defined as the proportion of pairs of genes assigned to the equivalent clusters over the total numbers of pairs exceeded 0.85 for $k \geq 5$ and we selected $k = 5$.

To determine enrichments of CTCF bookmarked and lost peaks in proximity to these clusters, we calculated Fisher Exact test $p$-values for the genes of a given cluster within $x$bp of a CTCF peak to a background of all genes clustered within $x$bp of a CTCF peak, for x in [1, 1e+6] bp.

Post-replication reactivation - repli-ATAC-seq repli-ATAC-seq reads were taken from GSE128643 (*Stewart-Morgan et al., 2019*). Paired end reads were trimmed by aligning read pairs to discover regions of reverse complementarity surrounded by Nextera sequencing adapters for ATAC-seq. Alignment and trimming was performed with the BioSequences package for Julia 0.6 (*Bezanson et al., 2017*). Reads were aligned to mm10 genome using Bowtie 2 (*Langmead and Salzberg, 2012*) with options '-k 10 -I 0 -X 1000 –no-discordant –no-mixed', and filtered for reads with a single alignment mean edit distance less than four between read pairs. For comparison to MINCE-seq we quantified the total number of cutsites from 0 to 100 bp repli-ATAC-seq fragments normalized to total 0–100 bp fragments per library, falling within +/- 100 bp of CTCF maximal motif in CTCF peaks, Esrrb maximal motif in Esrrb Peaks, Oct4/Sox2 maximal motif in Oct4/Sox2 peaks and the summit of p300 peaks. To determine groups with differing post-replication reactivation dynamics we took total cutsites of 0–100 bp repli-ATAC-seq fragments, normalized to total 0–100 bp fragments, within [−200, 0] bp of active promoters as determined by chromHMM ES cell data (*Pintacuda et al., 2017*), as used in *Figure 3D*. We filtered for promoters with normalized total accessibility over the [−200, 0]bp region of 0.15 in all samples and >0.5 in steady state. We employed a strategy identical to our analysis of post-mitotic gene reactivation. We normalized, nascent (Pulse), 30 min, 60 min and 120 min chase to steady state as $\log_2$ fold changes and clustered with k-means clustering. Similarly, we selected k = 5 as the first k for which the Rand Index exceeded 0.85 (see previous section). To determine enrichments of CTCF peaks in proximity to these clusters, we calculated Fisher Exact test $p$-values for the genes of a given cluster within $x$bp of a CTCF peak to a background of all genes clustered within $x$bp of a CTCF peak, for x in [1, 1e+6] bp.

## Acknowledgements

The authors acknowledge the Imagopole France–BioImaging infrastructure, supported by the French National Research Agency (ANR 10-INSB-04–01, Investments for the Future), for advice and access to the UltraVIEW VOX system. We also acknowledge the Transcriptome and EpiGenome, BioMics, Center for Innovation and Technological Research of the Institut Pasteur for NGS. We thank Steven Henikoff and Srinivas Ramachandran for advice on setting up MINCE-seq; Andrea Voigt for sharing protocols for EdU incorporation and click reactions; Shahragim Tajbakhsh for C2C12 and 3T3 cells; Marlies Oomen, Job Dekker and Gerd Blobel for discussions and critical reading of the manuscript. This work was supported by recurrent funding from the Institut Pasteur, the CNRS, and Revive (Investissement d'Avenir; ANR-10-LABX-73). EPN was supported by EMBO (ALTF523-2013), HSFP, and the Roddenberry Stem Cell Center at Gladstone. NO is supported by Revive. PN acknowledges financial support from the Fondation Schlumberger (FRM FSER 2017), the Agence Nationale de la Recherche (ANR 16 CE12 0004 01 MITMAT), the Ligue contre le Cancer (LNCC EL2018 NAVARRO) and the European Research Council (ERC-CoG-2017 BIND).

## Additional information

### Funding

| Funder | Grant reference number | Author |
| --- | --- | --- |
| Institut Pasteur | | Michel Cohen-Tannoudji Pablo Navarro |
| Centre National de la Recherche Scientifique | | Michel Cohen-Tannoudji Pablo Navarro |
| Agence Nationale de la Recherche | Investissement d'Avenir; Revive Labex; ANR-10-LABX-73 | Pablo Navarro |
| Agence Nationale de la Recherche | ANR 16 CE12 0004 01 MITMAT | Pablo Navarro |
| Ligue Contre le Cancer | LNCC EL2018 NAVARRO | Pablo Navarro |

| European Research Council | ERC-CoG-2017 BIND | Pablo Navarro |
|---|---|---|
| European Molecular Biology Organization | ALTF523-2013 | Elphège P Nora |
| Human Frontier Science Program | | Elphège P Nora |
| Schlumberger Foundation | FRM FSER 2017 | Pablo Navarro |

The funders had no role in study design, data collection and interpretation, or the decision to submit the work for publication.

### Author contributions

Nick Owens, Conceptualization, Data curation, Software, Formal analysis, Investigation, Methodology, Writing—original draft, Writing—review and editing; Thaleia Papadopoulou, Conceptualization, Formal analysis, Investigation, Methodology, Writing—original draft, Writing—review and editing; Nicola Festuccia, Investigation, Methodology, Writing—review and editing; Alexandra Tachtsidi, Validation, Investigation, Writing—review and editing; Inma Gonzalez, Sandrine Vandormael-Pournin, Investigation, Writing—review and editing; Agnes Dubois, Elphège P Nora, Benoit G Bruneau, Resources, Writing—review and editing; Michel Cohen-Tannoudji, Supervision, Investigation, Writing—review and editing; Pablo Navarro, Conceptualization, Supervision, Funding acquisition, Investigation, Methodology, Writing—original draft, Project administration, Writing—review and editing

### Author ORCIDs

Nick Owens https://orcid.org/0000-0002-2151-9923
Benoit G Bruneau https://orcid.org/0000-0002-0804-7597
Michel Cohen-Tannoudji http://orcid.org/0000-0002-6405-2657
Pablo Navarro https://orcid.org/0000-0002-2700-6598

### Ethics

Animal experimentation: All experiments were conducted according to the French and European regulations on care and protection of laboratory animals (EC Directive 86/609, French Law 2001-486 issued on June 6, 2001) and were approved by the Institut Pasteur ethics committee (n° dha180008).

### Decision letter and Author response

Decision letter https://doi.org/10.7554/eLife.47898.031
Author response https://doi.org/10.7554/eLife.47898.032

## Additional files

### Supplementary files

• Supplementary file 1. Describes all ChIP-seq, MNase H3 ChIP-seq and MINCE-seq samples sequenced in this study.
DOI: https://doi.org/10.7554/eLife.47898.019

• Supplementary file 2. Describes all CTCF and SMC1 binding regions identified in this study, their enrichments in interphase and mitosis and their bookmarking status.
DOI: https://doi.org/10.7554/eLife.47898.020

• Transparent reporting form DOI: https://doi.org/10.7554/eLife.47898.021

### Data availability

Sequencing data generated for this study have been deposited in GEO with accession GSE131356. Publicly available datasets used here: Festuccia et al. 2019; GEO accession: GSE122589; Teves et al. 2018; GEO accession: GSE109963; Stewart-Morgan et al. 2019; GEO accession: GSE128643.

The following dataset was generated:

| Author(s) | Year | Dataset title | Dataset URL | Database and Identifier |
|---|---|---|---|---|
| Owens N, Navarro P | 2019 | CTCF confers local nucleosome resiliency after DNA replication and during mitosis | https://www.ncbi.nlm.nih.gov/geo/query/acc.cgi?acc=GSE131356 | NCBI Gene Expression Omnibus, GSE131356 |

The following previously published datasets were used:

| Author(s) | Year | Dataset title | Dataset URL | Database and Identifier |
|---|---|---|---|---|
| Teves SS, Tjian R | 2018 | Role of TBP in reactivation of transcription following mitosis [RNA-Seq] | https://www.ncbi.nlm.nih.gov/geo/query/acc.cgi?acc=GSE109963 | NCBI Gene Expression Omnibus, GSE109963 |
| Owens N, Navarro P | 2019 | Transcription factor activity and nucleosome organisation in mitosis | https://www.ncbi.nlm.nih.gov/geo/query/acc.cgi?acc=GSE122589 | NCBI Gene Expression Omnibus, GSE122589 |
| Stewart-Morgan KR, Reverón-Gómez N, Groth A | 2019 | Transcription Restart Establishes Chromatin Accessibility after DNA Replication | https://www.ncbi.nlm.nih.gov/geo/query/acc.cgi?acc=GSE128643 | NCBI Gene Expression Omnibus, GSE128643 |

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
