## [Decision Letter]

Thank you for submitting your article "CTCF confers local nucleosome resiliency after DNA replication and during mitosis" for consideration by *eLife*. Your article has been reviewed by two peer reviewers, and the evaluation has been overseen by a Reviewing Editor and Jessica Tyler as the Senior Editor. The reviewers have opted to remain anonymous.

The reviewers have discussed the reviews with one another and the Reviewing Editor has drafted this decision to help you prepare a revised submission.

Summary:

The aim of this work is to resolve whether certain transcription factors have the ability to bookmark nucleosome ordering during and after chromatin disruptive phases of the cell cycle, the replication and the mitosis phases. This has important implication for the perpetuation of cell identities, at the scale of a cycling cell and also across generations of cells. By resetting chromatin to a naïve state, replication and mitosis may contribute to cellular plasticity; conversely, the ability to resist this or act immediately-early after replication or mitosis is likely to be towards the top of any hierarchical events that establish local epigenetic state. Acceptance for such 'chromatin bookmarking' is contentious in the field due to the limited resolution of most methodology to monitor TF dynamics in a temporal and locus-specific manner.

This study further expands on the authors' prior work identifying Esrrb as a bookmarking factor that maintains nucleosome organization at regulatory sequences across mitosis of embryonic stem cells. Using CTCF as a case example and further expanding to Esrrb, the authors put here forward the concept of "resilient transcription factor" by showing that:

1) CTCF binding is a strong determinant of nucleosome ordering, throughout the cell cycle;

2) CTCF relocates very rapidly after the passage of the disruptive replication fork, for a fast restoration of nucleosome order;

3) CTCF remains attached to the majority of its binding sites during mitosis, preserving nucleosome organization from mother to daughter cells. This stands in contrast with somatic cells, whereby CTCF loose sequence-specific binding on mitotic chromosomes, as shown by Dekker/Liu/Darzacq and also, the current authors.

The manuscript is concise and well argued, data are generally convincing and analyses carefully conducted, using convergent mapping methods for chromatin accessibility, nucleosome phasing and chromatin binding. Four TF binding sites were analyzed (CTCF, ESrrb, Oct4/ Sox4 and P300) and it is unclear whether resiliency- exhibited by two of them (CTCF or Esrrb)- is the norm or the exception. A larger catalog of TFs should be looked upon to draw conclusions and understand what it takes to be a bookmarking TF. Also, are TFs with resilient occupancy during mitosis systematically prone to rapid reestablishment post-replication? These general questions were raised by the reviewers but appear to be beyond the scope of the current paper. However, because the emphasis is mostly on CTCF, it is necessary to demonstrate that such resiliency has consequences on the continuity of expression programs across the cell cycle. Concern was also raised regarding the data with the AID-CTCF cells. Please address the major comments below in a revised version of the manuscript.

Essential revisions:

1) The authors propose the appealing idea that -in the context of ES cells- CTCF bookmarking may allow rapid restoration of gene activity after mitosis and hence maintenance of cellular identity during cell divisions, by keeping accessible the sites for immediate TF binding. It seems important to provide evidence for this, behind a mere description of persistent or rapid binding of CTCF, which may not have any biological function. Either comparison of transcript levels or kinetics of reactivation (likely nascent transcription) should be assessed before and after mitosis and compared between interphase CTCF sites that are bookmarked at mitosis and interface CTCF sites that are not. Or the CTCF-AID degradation system could also be used to address this point: around mitosis, are CTCF bookmarked sites affected in the expression reinitialization? Similarly, around replication, are CTCF binding sites affected in their levels/kinetics of expression? Choosing one or the other option, this could be assessed genome wide (polII occupancy, nascent RNA seq) or in a targeted manner, looking at a few loci with similar expression levels in pooled cells (RNA FISH, RT-qPCR). Conversely, if there are no functional consequences to losing mitotic bookmarking/nucleosome phasing, why do the authors think this is the case?

2) The authors indicate (and their data show) that AID-CTCF cells have decreased CTCF levels and a hypomorph phenotype, but that the phenotype is amplified in mitosis. This requires a bit more explanation, given that protein levels in mitosis do not appear different from interphase, based on Figure 3—figure supplement 1. Does the protein associate with chromosomes at levels similar to the wild type protein (i.e. by fractionation followed by Western blotting) in mitosis?

3) In Figure 6C-E, the full comparison of wild type and AID-CTCF, +/- IAA needs to be shown to illustrate how much of the loss of nucleosome phasing is mitosis-specific versus already present in the AID-CTCF cells in interphase. Please include statistics on the comparisons.

Preferred changes:

Analyses and Discussion:

1) Around 52 000 CTCF peaks were identified on bulk cells and around 29 000 in mitotic cells. It is assumed that these mitotic sites are all found among the 52 000 bulk sites, but it cannot be excluded that some mitosis-specific CTCF sites may exist, and would not have been seen from pooled cells. Figure 5C may suggest it is not the case, but this should be explicitly said.

2) Quality control data for the mitotic cells used for analysis (ES cells and other cell lines) should be included (H3S10p/DNA FACS profile or cell counts with a mitotic marker).

3) In comparing the MINCE-seq results with those from *Drosophila* S2 cells, it would be helpful to frame them in the context of cell cycle timing-at the 1 hour chase point. Where in the cell cycle are the labelled ES cells versus S2 cells?

4) Why are two different genome annotations (mm9 and mm10) used to analyze the data?

5) CTCF does not only act as a TF but also as a boundary factor for TADs, notably through Cohesin recruitment. TADs have been reported to dissolve in mitotic chromosomes (Dekker, Tanay groups) and the authors and other groups before (cited in the manuscript) reported that although the majority of CTCF ChIP peaks are maintained in mitotic cells, the interaction with Cohesin is lost. The authors should therefore discuss their findings In regards to cell cycle-regulated formation of TADs.

6) The authors should also discuss previously findings which argue that replication processes are inhibited by TF/Nucleosomes and that this contributes to an elevated mutation rate at CTCF sites (Reijns et al., 2015). These data are in agreement with CTCF binding early after replication and are independent of many of the technical issues that may arise from biochemical measurements.

---

## [Author Response]

[…] The manuscript is concise and well argued, data are generally convincing and analyses carefully conducted, using convergent mapping methods for chromatin accessibility, nucleosome phasing and chromatin binding. Four TF binding sites were analyzed (CTCF, ESrrb, Oct4/ Sox4 and P300) and it is unclear whether resiliency- exhibited by two of them (CTCF or Esrrb)- is the norm or the exception. A larger catalog of TFs should be looked upon to draw conclusions and understand what it takes to be a bookmarking TF. Also, are TFs with resilient occupancy during mitosis systematically prone to rapid reestablishment post-replication? These general questions were raised by the reviewers but appear to be beyond the scope of the current paper.

We thank the reviewers for their generally positive opinion of our manuscript. While we agree that it is of major importance to establish whether resiliency is an exception or the norm, we thank the Editor to understand this is not the main focus of the current paper, where we show for the first time that TFs such as CTCF can confer nucleosome resiliency to its genomic targets. The need to extend this observation to other TFs was already present in the Discussion (and in the last paragraph of the Discussion in the current version).

However, because the emphasis is mostly on CTCF, it is necessary to demonstrate that such resiliency has consequences on the continuity of expression programs across the cell cycle. Concern was also raised regarding the data with the AID-CTCF cells. Please address the major comments below in a revised version of the manuscript.

As explained below, we have made considerable efforts to raise the key point raised by the reviewers regarding the functional consequences of CTCF-mediated nucleosome resiliency. We have also experimentally consolidated the data with the AID-CTCF cells. We hope this improved version of our manuscript will now be positively considered by the reviewers and the Editor.

Essential revisions:1) The authors propose the appealing idea that -in the context of ES cells- CTCF bookmarking may allow rapid restoration of gene activity after mitosis and hence maintenance of cellular identity during cell divisions, by keeping accessible the sites for immediate TF binding. It seems important to provide evidence for this, behind a mere description of persistent or rapid binding of CTCF, which may not have any biological function. Either comparison of transcript levels or kinetics of reactivation (likely nascent transcription) should be assessed before and after mitosis and compared between interphase CTCF sites that are bookmarked at mitosis and interface CTCF sites that are not. Or the CTCF-AID degradation system could also be used to address this point: around mitosis, are CTCF bookmarked sites affected in the expression reinitialization? Similarly, around replication, are CTCF binding sites affected in their levels/kinetics of expression? Choosing one or the other option, this could be assessed genome wide (polII occupancy, nascent RNA seq) or in a targeted manner, looking at a few loci with similar expression levels in pooled cells (RNA FISH, RT-qPCR). Conversely, if there are no functional consequences to losing mitotic bookmarking/nucleosome phasing, why do the authors think this is the case?

We agree with the criticism that no evidence of functionality for resilient CTCF binding was provided in our manuscript. We report now (new Figure 7) correlative analyses using additional ES cell datasets (Teves et al., 2018 and Stewart-Morgan et al., 2019) that directly address the link between CTCF and transcription reactivation after mitosis and replication:

- The Tjian lab reported a genome-wide analysis of chromatin-associated RNAs, enriched in nascent pre-mRNAs, in mitosis and upon release in interphase. Using a clustering approach, we observe that genes displaying the fastest reactivation display a strong statistical enrichment in the vicinity of CTCF bookmarked sites (new Figure 7A, B). This enrichment is drastically reduced when considering sites that lose CTCF binding in mitosis. Therefore, these analyses indicate that CTCF bookmarking confers faster reactivation dynamics following mitosis.

- The Groth lab described repli-ATAC, a technique very similar to Mince-seq where chromatin accessibility is monitored immediately after replication and following different time-points post-replication, as the chromatin matures. First, we validated our observations: CTCF and Esrrb bound regions display faster accessibility reestablishment post-replication compared to Oct4/*Sox2* and p300 binding regions (new Figure 4—figure supplement 1B). Second, we focused on accessibility around Transcription Start Sites (TSSs), which generally reflect binding of the preinitiation complex of transcription. We observed that among the promoters displaying faster dynamics of TSS accessibility post-replication, CTCF targets were statistically enriched (new Figure 7C, D). This enrichment was in contrast not observed for TSSs displaying slow reactivation dynamics. Hence, echoing post-mitosis gene reactivation, CTCF may also confer fast promoter reactivation after replication.

We conclude, therefore, that CTCF binding is strongly associated with rapid reactivation of gene transcription after mitosis and replication, providing direct support to the functional role of resilient TF binding in gene transcription. In addition to new Figure 7, these analyses are thoroughly described at the end of the Results section and in the Discussion.

2) The authors indicate (and their data show) that AID-CTCF cells have decreased CTCF levels and a hypomorph phenotype, but that the phenotype is amplified in mitosis. This requires a bit more explanation, given that protein levels in mitosis do not appear different from interphase, based on Figure 3—figure supplement 1. Does the protein associate with chromosomes at levels similar to the wild type protein (i.e. by fractionation followed by Western blotting) in mitosis?

CTCF displays a hypomorph phenotype in interphase AID cells, possibly due to decreased levels but perhaps also because its fusion to GFP and AID may induce steric binding issues. In mitosis the chromatin changes considerably and the nuclear envelope disappears thereby reducing the effective concentration of TFs around the chromatin. In this context of unfavourable protein-DNA interactions, we believe it is not unconceivable that CTCF-GFP-AID fusion protein displays an increased hypomorph behaviour. We agree, however, that this was exclusively shown by ChIP-seq. As suggested by the reviewers we now provide a chromatin fractionation followed by Western blotting that fully confirms our observations: the CTCF-GFP-AID protein exhibits reduced chromatin association in interphase and even less in mitosis. This has been added as Figure 3—figure supplement 1B.

3) In Figure 6C-E, the full comparison of wild type and AID-CTCF, +/- IAA needs to be shown to illustrate how much of the loss of nucleosome phasing is mitosis-specific versus already present in the AID-CTCF cells in interphase. Please include statistics on the comparisons.

We understand the reviewer’s request and are now happy to provide in an updated Figure 6—figure supplement 1B, C, D the full range of raw comparisons within conditions and associated metrics and statistics. We hope the reviewers will agree that while already altered in interphase, the loss of nucleosome phasing in mitotic CTCF-Aid cells is major.

Preferred changes:Analyses and Discussion:1) Around 52 000 CTCF peaks were identified on bulk cells and around 29 000 in mitotic cells. It is assumed that these mitotic sites are all found among the 52 000 bulk sites, but it cannot be excluded that some mitosis-specific CTCF sites may exist, and would not have been seen from pooled cells. Figure 5C may suggest it is not the case, but this should be explicitly said.

We have so far never observed mitosis-specific peaks, although we agree with the reviewers that only a single cell analysis can fully discard this possibility. We have, however, robust evidence that at a small number of sites, CTCF binding is higher in mitosis than in interphase. These were visible in our original Supplementary Figure 4A, and enumerated in Materials and methods. To reiterate here, we observe 18,723 sites in which the interphase peak is larger than in mitosis, 10,456 sites where the peaks are comparable between interphase and mitosis and 360 sites in which mitosis peaks are larger than those in interphase. We now provide in Figure 5—figure supplement 1 more explicit descriptions of these categories and matching nucleosome profiles that agree with our conclusions.

2) Quality control data for the mitotic cells used for analysis (ES cells and other cell lines) should be included (H3S10p/DNA FACS profile or cell counts with a mitotic marker).

We routinely count mitotic cells by microscopy using DNA staining, this is in our hands the more reliable quantification. FACS profiles can in fact be misleading because late G2 cells are also highly positive for typical markers such as H3S10p or MPM2: only a direct visualisation of mitotic figures can be used as a rigorous criterion. As stated in our Materials and methods, all samples with more than 5% contaminants using stringent visual criteria are discarded.

Moreover, we would like to highlight that the contamination with interphase cells is obviously important (we have done significant efforts in the past to address this issue quantitatively – see Festuccia et al., 2019). However, the chromatin used here to monitor CTCF binding in mitotic wild-type ES cells was previously used in studies where TFs other than CTCF displayed no binding in mitosis (Festuccia et al., 2019), ruling out a potential contamination issue. This is further supported by the full loss of Cohesin we observe and report here in parallel ChIP-seq. Moreover, the existence of our CTCF binding categories showing differential levels of bookmarking (Figure 5—figure supplement 1) also imply that the observed mitotic binding is not due to interphase contamination. In the new cell lines analysed here (CTCF-AID ES cells, 3T3 and C2C12 fibroblasts), we observe no bookmarking, so the <5% contamination does not lead to false discovery.

3) In comparing the MINCE-seq results with those from *Drosophila* S2 cells, it would be helpful to frame them in the context of cell cycle timing-at the 1 hour chase point. Where in the cell cycle are the labelled ES cells versus S2 cells?

Given the length of the S phase in the two cell types (around 15h in S2 cells and 8h in mouse ES cells, the vast majority of labelled cells (in the pulse of after 1h chase) are still in S phase.

4) Why are two different genome annotations (mm9 and mm10) used to analyze the data?

All analyses have been performed against mm10. There was a typographical error in our Materials and methods, which has now been corrected.

5) CTCF does not only act as a TF but also as a boundary factor for TADs, notably through Cohesin recruitment. TADs have been reported to dissolve in mitotic chromosomes (Dekker, Tanay groups) and the authors and other groups before (cited in the manuscript) reported that although the majority of CTCF ChIP peaks are maintained in mitotic cells, the interaction with Cohesin is lost. The authors should therefore discuss their findings In regards to cell cycle-regulated formation of TADs.

We now discuss more directly CTCF bookmarking and the loss of Cohesin in light of the loss of TADs in mitosis, as requested by the reviewer (Discussion, last paragraph).

6) The authors should also discuss previously findings which argue that replication processes are inhibited by TF/Nucleosomes and that this contributes to an elevated mutation rate at CTCF sites (Reijns et al., 2015). These data are in agreement with CTCF binding early after replication and are independent of many of the technical issues that may arise from biochemical measurements.

We sincerely thank the reviewers for this interesting comment. This notion has now been included in the first paragraph of our Discussion.